# Historical snow measurements in the Central and Southern Apennine Mountains: climatology, variability and trend

Vincenzo Capozzi[1], Francesco Serrapica[1], Armando Rocco[1], Clizia Annella[2,1], Giorgio Budillon[1]

[1]Department of Science and Technology, University of Naples "Parthenope", Isola C4, CAP 80143, Italy
[2]Center of Excellence for Telesensing of Environment and Model Prediction of Severe events, University of L'Aquila, L'Aquila, Italy

*Correspondence to*: Vincenzo Capozzi (vincenzo.capozzi@uniparthenope.it)

**Abstract.** This work presents an analysis of historical snow precipitation data collected in the period 1951-2001 in Central and Southern Apennines (Italy), an area scarcely investigated so far. To pursue this aim, we used the monthly observations of the snow cover duration, number of days with snowfall and total height of new snow collected at 129 stations located between 288 and 1750 m above the sea level. Such data have been manually digitized from the Hydrological Yearbooks of the Italian National Hydrological and Mareographic Service. The available dataset has been primarily analyzed to build a reference climatology (related to 1971-2000 period) for the considered Apennine region. More specifically, using a methodology based on Principal Component Analysis and k-means clustering, we have identified different modes of spatial variability, mainly depending from the elevation, which reflect different climatic zones. Subsequently, focusing on the number of days with snowfall and snow cover duration on the ground, we have carried out a linear trend analysis, employing the Theil-Sen estimator and the Mann-Kendall test. An overall negative tendency has been found for both variables. For clusters including only stations above 1000 m above the sea level, a significant (at 95% confidence level) decreasing trend has been found in winter season (i.e. from December to February): −3.2 [−6.0 to 0.0] days/10 years for snow cover duration and −1.6 [−2.5 to −0.6] days/10 years for number of days with snowfall. Moreover, in all considered seasons, a clear direct relationship between trend magnitude and elevation has emerged. In addition, using a cross wavelet analysis, we found a close in-phase linkage on decadal time scale between the investigated snow indicators and the Eastern Mediterranean Pattern. For both snow cover duration and number of days with snowfall, such connection appears to be more relevant in full (i.e. from November to April) and in late (i.e. from February to April) seasons.

## 1 Introduction

In recent years, a great deal of attention has been devoted to the study of past snow variability worldwide, mainly in mountain regions. The great interest in this crucial climate variable is motivated by several reasons. The snow, in fact, is a pivotal component of the hydrological cycle and exerts, at the same time, a relevant impact on the energy balance, controlling the land

surface albedo. In addition, the snow strongly affects the complex ecosystems of mountain areas, as well as the biogeochemical
cycles (e.g. Magnani et al., 2017). Lastly, the occurrence as well as the persistence of snow on the ground is decisive for winter
tourism and for several economic activities (for instance, hydropower production). Therefore, considering also the recent
climate changes, that are posing serious threats on the cryosphere and mountain regions (e.g. Mote et al., 2018; Kotlarski et
al., 2022), it is crucial to recover and analyse historical long-term time series of snow data to assess its variability and
tendencies.

In last decades, the satellite observations and the climate model reanalyses have offered new opportunities of build-up snow
climatologies worldwide (e.g. Bormann et al., 2018; Olefs et al., 2020), especially in regions in which the availability of in
situ data is scarce or totally absent. However, it should be noted that remote sensing could provide information about snow
cover at an adequate resolution for reliable climatological analyses only for the last 20 years, after the deployment of the
Moderate Resolution Imaging Spectroradiometer (MODIS) constellation (Fugazza et al., 2021; Gascoin et al., 2022; Dumont
et al., 2024). On the other hand, as highlighted by Vernay et al. (2022), the reliability of reanalyses data can be adversely
affected by the low spatial resolution and by the rough representation of several sub-grid processes, such as the orographic
precipitation and the local thermal inversion in mountain valleys. The latter can strongly condition local nivometric regimes,
particularly in mountainous areas characterized by complex orography. For such reasons, despite their well-known weaknesses
(Notarnicola, 2020), the ground-based historical observations can be still considered a cornerstone for studies searching for
evidence of past snow variability. Accurate in situ snow measures open the chance to analyse in depth the climate change
impacts in mountain areas and their relationship with the altitude, especially when they can be coupled with high quality and
homogenized temperature and total precipitation measurements (Beaumet et al., 2021). In addition, the ground observations
can be considered an invaluable benchmark to validate satellite and modelled snow data.

In the large body of available scientific literature, snow has been investigated employing different parameters, namely the
snow depth (hereafter, HS), the height of new snow (hereafter, HN), the snow water equivalent, the snow cover area, the snow
cover duration (hereafter, SCD) and the number of days with snowfall (hereafter, NDS). Note that in literature, SCD generally
indicates the number of days with snow cover on the ground during a given period, whereas the NDS parameter represents the
number of days, in a determined time interval, in which the amount of fresh snow reaches a determined threshold (usually, 1.0
cm).

Focusing on the Central Mediterranean region, which can be considered a key-area for study of climate changes mainly due
its complex meteorological regime and to its challenging topography, a lot of research has been paid to the Alpine region
(Marty, 2008; Terzago et al., 2010; Valt and Ciafarra, 2010; Marty and Blanchet, 2012; Scherrer et al., 2013; Terzago et al.,
2013; Marcolini et al., 2017b; Matiu et al., 2021; Colombo et al., 2022; Bertoldi et al., 2023; Colombo et al., 2023). In this
area, in fact, there is a great availability of snow climatological time series, some of them stretching back to the late 18th century
(Leporati and Mercalli, 1994). Although it is clearly challenging to draw a coherent picture regarding changes in Alpine
nivometric regimes, since trends direction and significance are highly dependent on the considered time period, a general

decreasing tendency has been found for SCD in the period 1972-2006 (Bartolini et al., 2010) as well as in HS for the period 1971-2019 (Matiu et al., 2021).

The Apennine region, despite having a good heritage of old snow data, has been poorly investigated up to now. The peer-reviewed literature for this area counts only few recent works (Petriccione and Bricca, 2019; Diodato et al., 2022; Capozzi et al., 2022; Annella et al., 2023), which presented evidence based on a single or few climatological time series. These studies highlighted a general negative tendency in snow for different indicators, except for the last 20 years, in which a recovery of SCD, HN and NDS has been detected in the Southern Apennines (Capozzi et al., 2022; Annella et al., 2023). Other interesting results have been provided by two reports (Fazzini et al., 2006; Fazzini, 2007) published in the official information magazine of the Interregional Association for coordination and documentation of snow and avalanche problems (https://aineva.it/en/neve-e-valanghe-magazine/, last access on 16/01/2024). More specifically, Fazzini et al. (2006) analysed the 1982-2004 period, finding, for the Apennines area, a marked spatial heterogeneity in HN, SCD and NDS trends. From this study, in fact, emerged a strong positive tendency for the investigated variables in the Northern Apennines (eastern sector), a negligible trend in Central Apennines and local positive tendencies in the southern sector. Fazzini (2007) has obtained a similar result for the number of days with HN > 5 cm.

The peripheral attention dedicated to the Apennines can be mainly attributed to the very fragmented management of the meteorological monitoring network, which resulted into a non-uniform spatial and temporal coverage of snow data.

Historically, the snow monitoring in the Apennines areas has been handled by the Italian National Hydrological and Mareographic Service (hereafter, NHMS). The latter managed the hydro-meteorological data collection in Italy from 1917 to 2002 and was structured into 14 different compartments (Parma, Venezia, Genova, Bologna, Pisa, Rome, Pescara, Naples, Bari, Catanzaro, Palermo, Cagliari, Trento and Bolzano), defined based on the water catchment areas of the main Italian rivers. The NHMS snow dataset is currently not available into an easily accessible digitized format and, therefore, it has been largely unexploited so far.

After the disposal of NHMS, whose competences were transferred to the local regional agencies according to the new Legislative Decree issued by Italian government, many historical stations were dismissed or relocated. Unfortunately, the monitoring of snow precipitation was interrupted, except in few areas, mainly in the Emilia-Romagna and in Abruzzo regions, where automatic nivometric stations have been progressively installed (Tecilla, 2007). Additional contributions to the snow monitoring in Apennines Regions came from the Meteomont Service, which is managed by the Italian Arm of Carabineers, and from the Meteorological Service of the Italian Air Force (hereafter, MSIAF). The Meteomont network started in 1980s for avalanches danger assessment on synoptic/regional scale and actually consists, for Apennines, of 84 manual stations and 2 high altitude surveys (https://meteomont.carabinieri.it/stazioni-manuali?lang=en, last access on 04/01/2024). The data collected by such stations are publicly available in a digitized format through the Meteomont website (https://meteomont.carabinieri.it/archivio-condizioni-meteonivologiche?lang=en, last access on 04/01/2024). However, most of the time series are strongly incomplete and, therefore, their use for climatological purposes is challenging or prohibitive.

The MSIAF snow measurements are available since 1981 for 80 monitoring sites and consist of daily observations of HS and

of three-hourly measurements of the snow water equivalent (Fazzini et al., 2006). The entire Apennines region is monitored by MSIAF network through 15 stations, having an altitude between 352 and 2165 m above sea level (hereafter, ASL). However, such data are not publicly accessible and, more importantly, are strongly unevenly distributed in space and altitude, so they are not suitable for a reliable climatological characterization of the Apennines region.

105 In the light of this state of the art, a relevant lack of research exists in the knowledge of past snow variability in the Apennines. This works aims to provide a contribution to fill this gap, through the rescue of 281 historical time series of snow data collected by NHMS network in an area including a large part of the Central Apennines and a small sector of the Southern Apennines. After a careful quality control, a complete and high-quality dataset consisting of 110 and 114 monthly time series of SCD and NDS, respectively, collected during the period 1951-2001 and of 120 monthly time series of HN measured in the 1971-2001

110 period has been obtained. This dataset has been adopted to accomplish the two main goals of this study:

- Building-up an updated and solid reference climatology for SCD, NDS and HN variables (related to 1971-2000 period) for the considered Apennine region.
- Providing new evidence about long-term tendencies in NDS and SCD for the study area, analysing their relationship with the elevation.

115 The remainder of the paper is organized as follows: Section 2 describes the study area, the data and the methods, Section 3 presents the results, Section 4 is dedicated to the discussions and, finally, Section 5 provides the conclusions.

## 2 Materials and methods

### 2.1 Study area

120 The Apennines Mountains consist of parallel mountain ranges extending for about 1200 km from northwest to southeast along the length of the Italian Peninsula. They are conventionally subdivided into three different sectors: the northern sector, including the Ligurian and the Tuscan-Emilian Apennines, the central sector, encompassing the Umbria-Marche Apennines and the Abruzzi Apennines, and the southern sector, comprising the Samnite and Campania Apennines, the Lucan Apennines and the Calabria and Sicily Apennines. In this study, we focused on a study region embracing a large portion of the Central

125 Apennines and a small sector of the Southern ones (Fig. 1). This area extends from 40.5 to 43.5°N in latitude and from 12.5 to 16.0°E in longitude and includes the following Italian administrative regions: Umbria, Lazio, Abruzzo, Molise, Campania, Puglia and Basilicata. It has a very complex orography, consisting of several mountain ranges. The main orographic features, highlighted by filled-in brown triangles in Fig. 1, are (from north to south): the Sibillini mountains (2476 m ASL), the Laga mountains (2458 m ASL), the Reatini mountains (2217 m ASL), the Gran Sasso area (2912 m ASL), the Sirente-Velino

130 mountains (2487 m ASL), the Majella massif (2793 m ASL), the Marsicani mountains (2285 m ASL), the Matese massif (2050 m ASL), the Partenio (1598 m ASL), the Picentini mountains (1809 m ASL) and the Vulture-Li Foj area (1365 m ASL). Moreover, the study area also includes several Apennine offshoots, marked as filled-in green triangles in Fig. 1, such as the

Lazio Sub-Apennines (2063 m ASL), the Daunian mountains (1132 m ASL), the Gargano massif (1065 m ASL), and the Murge plateau (686 m ASL). The study region is bounded by flat areas which gradually slope down to the Tyrrhenian Sea (to the west) and to the Adriatic Sea (to the east).

The climate of this area presents distinct Mediterranean features, with a precipitation maximum between late autumn and mid-winter and a relevant minimum in mid-summer. According to Crespi et al. (2018), the spatial distribution of the accumulated precipitation is strongly conditioned by the orography of the region and exhibits a marked west-to-east gradient, with drier conditions along the Adriatic sector. It is interesting highlighting that, on average, the highest annual precipitation amounts (up to 2200-2500 mm) are observed in the massifs of the Southern Apennines, i.e. in the Matese, Partenio and Picentini areas. Although they have a lower altitude than the mountains of the Central Apennines, such reliefs lie in a relatively less-complex orographic context and, therefore, they receive precipitation by a wide spectrum of synoptic patterns (Capozzi et al., 2022). Regarding the snow precipitation, the only climatological reference is the old study of Gazzolo and Pinna (1973). This work provided a coarse climatology for the HN, SCD and NDS parameters for the whole Italian Peninsula, using the data collected by NHMS during the 1921-1960 period. According to Gazzolo and Pinna (1973), the major peaks of the Central Apennines (Gran Sasso, Sibillini, Laga Mountains and Majella) received, in the considered time interval, up to 400 cm of fresh snow per year. In the mountainous areas of the Southern sector (Matese, Partenio and Picentini), the average total yearly HN is slight above 200 cm. In contrast to what is generally observed for the total precipitation, the snowfall amounts observed in the eastern slopes of the Central Apennines and in adjacent flat and coastal areas are higher than those measured in the western sectors. This can be related to the effects of the cold continental air masses coming from Balkan region and Eastern Europe, which stimulate abundant snowfall precipitation through two main mechanisms: the vertical transport of moisture and heat connected to their passage over the Adriatic Sea, and the orographic forcing, which is related to their interaction with the mountain ranges. Regarding the SCD, the yearly climatological value is between 50 and 100 days (or greater) in the main reliefs of the Central Apennines, whereas it is generally below 50 days in the Southern massifs. The frequency of occurrence of snowfall events is very high (25-50 days) in the Central sector, while, according to Gazzolo and Pinna (1973) it is lower than 20 days in the Southern areas.

The mean annual temperature exhibits a strong altitudinal gradient, decreasing from the 16-17°C of the coastal areas to the 13-14°C of the base of Apennines and, finally, to the 2-4°C of the highest peaks of the Central sector (Curci et al., 2021). As perfectly testified by the climatology presented in Brunetti et al. (2014), the valleys and the sub-mountain areas of the Abruzzo region have a mean annual temperature lower than the most of hilly regions of Campania, Puglia and Molise. This difference is mainly due to the minimum temperature (Curci et al., 2021) and can be ascribed to the frequent occurrence, in the Abruzzo valleys, of the thermal inversion phenomenon.

## 2.2 Data rescue

In this study, we have exploited the database of four NHMS compartments: Naples, Bari, Rome and Pescara. The data collected by the stations belonging to the NHMS network were published in the Hydrological Yearbooks, which are freely accessible in

printed version (i.e. as scanned images in portable document format) through the Italian Institute for Environmental Protection and Research (ISPRA) website (http://www.bio.isprambiente.it/annalipdf/, last access on 09 January 2024). The editing and the publishing of the Hydrological Yearbooks were handled by the Departmental Office of the NHMS responsible for a specific compartment. Each Hydrological Yearbook contains the data collected in a certain year and was generally structured in two

different parts: the Part I, which includes the thermometric and pluviometric measurements and the Part II, which contains a wide spectrum of data, related to precipitation, hydrology, groundwater levels, exceptional events and tide measurements (https://www.isprambiente.gov.it/en/projects/inland-waters-and-marine-waters/hydrological-yearbooks, last access on 09 January 2024). The snow data are included in the Part II of the Hydrological Yearbooks from 1917 to 1934 and in the Part I from 1935 onwards.

More specifically, the snow data are reported for October to May period and consist of the following parameters: SCD, NDS, HS and HN. From 1917 to 1934, the available measurements include the daily HN, the corresponding snow water equivalent amount, the monthly NDS value and the HS value before the occurrence of a determined snowfall event. Subsequently, the snow data are reported with a different format. Regarding the SCD and NDS parameters, monthly data are available from 1935 to 1999 for Naples and Rome compartments, from 1935 to 2000 for Bari compartment and from 1935 to 2013 for Pescara

compartment. The temporal coverage of HS data resembles the one of SCD and NDS; however, it should be highlighted that for this parameter only three daily observations per months (at the end of each decade) are available from 1935 to 1971 and only one (in the last day of a determined month) in the subsequent periods. As concerns the HN variable, unfortunately no data are available from 1935 to 1970, whereas monthly observations are reported from 1971. The snow measurements have been manually performed using a traditional nivometer, consisting of a snowboard and a graduated yardstick (De Bellis et al., 2010).

It is important to highlight that according to the NHMS standard, the monitored snow parameters are defined as follows:

- SCD is the total number of days in a given month or in a given season with snow depth on the ground >=1 cm.
- NDS is the total number of days in a given month or in a given season on which the accumulated snowfall (i.e. the amount of fresh snow with respect to the previous observations) is at least 1 cm.
- HN is the daily or monthly amount of fresh snow (expressed in cm). The monthly value is intended as the sum of
daily HN data observed in a determined month.
- HS is the daily snow depth on the ground (expressed in cm).

From 1917 to the end of 1940s, the data availability is limited and is strongly conditioned by the First and Second World Wars period (in which many stations temporarily interrupted their monitoring activity). The number of stations reporting snow data

increased in the early 1950s and it was fairly stable until the end of 1990s except for some isolated drops (in 1989 and in 1997). Subsequently, after the closure of NHMS, the data availability strongly decreased and was restricted to some stations of the Abruzzo region, which continued to collect snow data under the management of local regional authorities, and to the Montevergine Observatory station (Southern Apennines), which autonomously continued the meteorological parameters recording (Capozzi et al., 2020).

In this study, we decided considering the 1951-2001 period, which corresponds to the years with the highest number of data. More specifically, we have digitized the SCD, NDS, HN and HS data collected by stations having an elevation greater than 250 m ASL In other words, we have excluded the stations located in flat areas or along the coasts, in which the occurrence of snow is relatively rare. Using this criterion, we have retrieved 281 stations, having an elevation ranging between 288 and 2125 m ASL Note that for the digitization process we have used a simply "key-entry" method. Despite the recent introduction of

new approaches and methodologies based on the optical character recognition software and machine learning tools, the manual transcription is still the most accurate technique for climate data digitization. As shown by Brönnimann et al. (2006), the manual method has the lower error rate and well fits the recommendations and the standards practises of the World Meteorological Organization (WMO, 2016), although it is the slower in terms of amount of rescued data per unit of time. The digital templates have been developed in Microsoft Excel and have been structured into different spreadsheets, one dedicated

to the station metadata and the other ones to the data of the available snow parameters.

A complete list of all rescued stations, with details about geographical coordinates (latitude, longitude and height ASL), membership NHMS compartment and percentage of available data, is provided in the Supplementary Material.

## 2.3 Data, quality control and homogenization

In this work, we have analysed the SCD and NDS data collected in the 1951-2001 period and the HN data measured between 1971 and 2001. We have decided not to consider the HS data: the latter, in fact, have been reported in the Hydrological Yearbooks with a format, consisting in three or one daily observations per month, which is not suitable for a reliable climatological analysis.

Fig. 2 presents the histograms of the data availability for three considered parameters, SCD (Fig. 2a), NDS (Fig. 2b) and HN

(Fig. 2c). The data availability has a clear bimodal distribution, with two distinct peaks, one between 0 and 10% and the other between 85 and 100%. From a simple visual inspection of Fig. 2, it emerges that a non-negligible fraction of the rescued stations has a limited data availability. For SCD and NDS (HN), the 39% (37%) of the measuring stations has a data availability less than 50%. According to the criteria suggested by the World Meteorological Organization (WMO, 2008), we have discarded the stations with less than 80% of the available data in the observation period. Moreover, we have rejected some

stations belonging to the Rome compartment, namely Abeto, Castelluccio di Norcia, Monte Terminillo (only for SCD and NDS) and Bagnara (only for SCD), due to the presence of many suspicious records. This screening yielded a subset consisting of 129 stations, whose position is shown in Fig. 3a. The spatial distribution of the stations is quite uniform over the entire region, except for the northern side (Umbria-Marche Apennines): for this area, there are only four stations located at an altitude ranging between 529 and 750 m ASL. The density of stations is particularly high in the proximity of the main mountain ridges

of Abruzzi and Samnite Apennines. In the Southern sectors, only one station (Montevergine Observatory) is located above 1000 m ASL. Regarding the elevation distribution, which is sketched in Fig. 3b, a relevant number of stations (69) is between

600 and 900 m ASL, 27 stations are below 600 m ASL and the remaining (33) are above 900 m ASL; the elevation ranges from a minimum of 288 m to a maximum of 1750 m ASL.

The considered dataset has been subjected to an accurate quality control (hereafter, QC). It is widely known, in fact, that the quality assurance of climate data is crucial to improve the confidence in any further analysis. As pointed out in many papers, the reliability and the consistency of an historical climatological time series can be affected by several artefacts and errors, caused by instruments failures, human mistakes in data collection and inaccuracies in the digitization of paper-based data. In this study, we have developed a QC strategy consisting of three statistical tests:

- The gross error test, which flags the data that are above or below acceptable physical limits;
- The consistency test, which involves an inter-variable check;
- The tolerance test, which is focused on the outlier detection.

Note that this QC strategy has been applied to the monthly SCD and NDS time series available in the 1951-2001 period and to the monthly HN time series available in the 1971-2001 time segment.

The gross error test aims to identify the clearly erroneous values and consists of comparing the monthly SCD, NDS and HN values to their physical limits. For SCD, we have checked for cases in which, for a determined month and for a certain station, the number of days with snow on the ground is greater than the number of days in that month (e.g. SCD is 32 days in January). For NDS, we have applied a very similar criterion, flagging as "gross errors" the circumstances in which the NDS value is equal or greater than the number of days in a determined month. According to this criterion, the instance in which a new snowfall event occurred in every day of a certain month is considered an implausible situation. For monthly HN values, we have considered the limit (500 cm) recommended by WMO (2008).

The second step of the QC process aims to detect inconsistencies between pairs of the investigated variables. More specifically, the purposes of this test is to detect the following instances: i) the NDS value for a determined month and a certain station is greater than the SCD value; ii) the NDS value for a determined month and a certain station is zero and the HN value is greater than zero; iii) the NDS value for a certain month and a determined station is greater than zero and the NH value is null. By applying the gross and the consistency checks, 1527 monthly invalid data were found: in particular, the 2.0% of the erroneous data emerged from the gross test, whereas the remaining 98.0% represents the outcome of the consistency test. A relevant fraction of the bad data can be attributed to instances in which the NDS value is greater than the SCD value. In some Hydrological Yearbooks, in fact, the NDS value is reported in the column dedicated to the SCD value and vice versa. Therefore, in most of the cases, the identified errors have been easily corrected; in other circumstances, they have discarded and replaced with a missing data marker.

The tolerance test has been performed using the Climatol method. The latter has been developed by Guijarro (2018) and is widely employed for the QC, homogenization and in filling of the missing data for a set of climatological time series. The Climatol data processing starts with a normalization of the original data. In this respect, Climatol offers different approaches for normalization, depending on the climatological variable. In this study, the type of normalization (*std*) has been set to 1 (which means that data normalization is based on deviations from mean) for SCD and NDS, whereas we selected *std* = 2

(which means normalize using ratio to normal climatological value) for HN. The approach used by Climatol to detect outliers is inspired by the principles of the spatial consistency check. In particular, for any candidate time series, this method uses data from neighbouring stations to estimate a corresponding reference series as a weighted average, employing a geographic proximity criterion using Euclidean distances. In the default settings of the Toolbox, the vertical and horizontal distances

(expressed in meters and kilometres, respectively) between a suitable neighbouring station and the candidate one have the same weight. Following Buchmann et al. (2022), to take into consideration the influence of altitude on the snow, in this study we have adjusted the scale parameter of the vertical coordinate ($wz$) so that the elevation counts 100 time more; in other words, the approach used in our work means that an altitude difference of 500 m corresponds to a horizontal distance of 50 km. The estimated reference series are used to create time series of anomalies for their corresponding observed series by subtracting

the estimated values from the observed ones. The values of the anomalies time series that exceed a determined threshold ($dz.max$) are labelled as outliers and so the correspondent data in the original series are discarded. More specifically, the $dz.max$ value, set by default to ±5 standard deviations, was properly tuned to ensure that the flagged outlying values were not rejected because of their extremeness. After several sensitivity experiments, in which we manually inspected the data flagged as potential outliers, the $dz.max$ parameter has been set as follows: $dz.max = 15$ for SCD and NDS and $dz.max = 20$ for HN. Using

this criterion, the tolerance test flagged as outliers only two NDS monthly observations, related to Frigento and Roccasicura time series.

Climatol has been employed in this study also to check for homogeneity of the investigated time series. The use of this toolbox for the homogenisation of snow data has been explored, with encouraging results, in some recent works (Buchmann et al., 2022; Buchmann et al., 2023). As described in detail by Guijarro (2018) and by Kuya et al. (2022), the Climatol

homogenization method is based on the Standard Normal Homogeneity Test (SNHT; Alexandersson, 1986) for the identification of the breaks and on a linear regression approach for the adjustments (Easterling and Peterson, 1995). The SNHT falls within homogenization procedures that are able to identify an inhomogeneity without knowing a priori the time of the break point in the time series and that can also estimate the magnitude of the detected break. The basic idea underlying this method consists in using neighbouring stations as a reference to identify inhomogeneities in the station being tested (the

candidate station). Such assumption requires the existence of a sufficient correlation level between test and reference stations. More specifically, SNHT uses normalised series of the ratios/differences (hereafter, $Q$) between e.g. precipitation/temperature at candidate station and neighbouring reference stations. The test is based on the null hypothesis that the $Q$ series has a constant mean level, i.e. that the candidate series is homogeneous, and the alternative hypothesis that the mean level of the $Q$ series changes abruptly from one level to another at some time. For each point of the time series, a test value, based on a comparison

between the means of the two subsamples before and after the potential breakpoint, is computed as described in detail in Alexandersson and Moberg (1997). The null hypothesis is rejected if the maximum test value of all dividing points in the $Q$ series is greater than a predefined critical level. In Climatol, the SNHT is applied to the anomalies time series previously introduced in the description of the tolerance test. In brief, the Climatol homogenization process is structured in two procedures: the application of the SNHT on stepped overlapping temporal windows and on the whole series. In the first one,

called "stepped overlapping windows", the toolbox computes the SNHT test for all series, retaining the maximum SNHT value for each series. The series having a maximum SNHT value greater than a specific threshold (*snht1*) are split into two subseries at the point of the maximum SNHT value. Subsequently, the sub-series are tested again and the procedure is iterated until the maximum SNHT value of the sub-series is below the *snht1* threshold. After this procedure, the test is applied to the whole series in order to detect further breaks, using a threshold value *snht2*. Once detected a break for a determined candidate time series, the latter is corrected back in time starting from the most recent homogeneous time interval. The break magnitude corrections are computed as the variation of the mean before and after homogenisation procedure. More specifically, given a time series *Y*, the correction factor (*CF*) is calculated as:

$$CF = \frac{\sigma_Q\, Y_b + \, Q_m}{\sigma_Q Y_a + \, Q_m}$$

(1)

Where $Y_b$ and $Y_a$ are the mean values between the beginning of the measurements of *Y* and the break point (before) and from the break point to the end (after), respectively. *Q* is the non-standardised ratio time series, defined as the ratio between the reference and candidate, and $\sigma_Q$ and $Q_m$ are the standard deviation and mean of *Q*, respectively.

Additional details about the calculation of the adjustment factor can be found in Guijarro (2018), in Kuya et al. (2022) and in Buchmann et al. (2023). The last step of Climatol processing consists in the filling of all missing values using the weighted ratios of neighbouring series and in the production of the final high quality, homogeneous and complete time series. It is important highlighting that Climatol offers the opportunity to carry out a first explanatory analysis of the data, which is very useful for the tuning of several parameters, including *snht1* and *snht2*. The main settings adopted to run Climatol for tolerance test of QC and homogenization are listed in Table 1.

Using this set-up, Climatol flagged as inhomogeneous seven SCD and two NDS time series. Details about date in which the breaks occurred and the corresponding value of SNHT are supplied in the Supplementary Material. From a visual inspection of such time series, the results of the homogeneity test seemed very reasonable. The identified breaks were further examined against the metadata reported on the Hydrological Yearbooks. However, the latter contain only few useful information, that allowed to verify only if the stations were relocated (this is not the case for any of the stations identified as inhomogeneous). We therefore do not have enough information to determine the cause of the inhomogeneities. We decided to adopt a precautionary approach and, therefore, the detected breaks were accepted.

**2.4 Cluster analysis**

In order to building up a reference climatology for the three parameters investigated in this study, the SCD, NDS and HN time series have been grouped into different clusters, each representing a specific climatic zone. Following the previous literature, we used a multivariate method based on the Principal Component Analysis (PCA) and Cluster Analysis (CA). This approach

has been employed in different areas (Kidson, 1994; Sumner et al., 1995; Fragoso and Tildes Gomes, 2008; Capozzi et al., 2023), proving to be reliable in the classification of meteorological data.

To search for dominant spatial patterns, the (non-rotated) PCA has been applied in T-mode to a dataset consisting of $n$ «observations» (i.e. the stations, 110 for SCD, 114 for NDS and 120 for HN) and $m$ «variables» (i.e. the monthly data, 612 for SCD and NDS and 372 for HN). It is important highlighting that the PCA has been employed not as a merely data reduction technique, but as a method which guarantees that only the fundamental modes of variation of the data are taken into account. Using the scree plot, we have determined the appropriate number of principal components (PCs) for each of the three investigated parameters. After this pre-processing, the well-known non-hierarchical k-means method (Anderberg, 1973) was then performed using the selected PCs scores as input. In the selection of the optimal number of clusters, we searched for the best trade-off between subjective evaluation, based on the patterns expected from the climatic and topographic drivers, and the semi-objective metrics («elbow method»).

## 2.5 Statistical analyses

In order to evaluate the magnitude of the trend slope in SCD and NDS time series, the well-known Theil-Sen non-parametric test was employed (Sen, 1968). This procedure is largely used in the hydro-meteorological framework because of its robustness in the presence of outliers in the series (Song et al., 2015; Bartolomeu et al., 2016; Ortiz-Gomez et al., 2020). To assess the trend magnitude, the Theil-Sen procedure estimates the trend slope in a determined sample for all data pairs. The median of all samples computed for each data pairs coincides with steepness of the trend. On the other hand, the statistical significance was assessed with the Mann-Kendall non-parametric test (Mann, 1945; Kendall, 1962). This test is a rank-based method for assessing the presence of increasing or decreasing monotonic trends in time series data and it is often used because of its property of requiring minimal assumptions about the data that need to be tested (Hamed, 2008). The significance levels of 0.05 and 0.1 (i.e., the 95% and 90% confidence levels, respectively) have been used to test the null hypothesis that there is no trend in the data. It is important to highlight that prior to the application of these statistical tests, the pre-whitening technique was used to reduce the effect of lag-one autocorrelation in the analyzed time series (e.g. Caloiero et al., 2011, Tramblay et al. 2013). In addition, to measure the relationship between the linear trend of the analysed parameters and the elevation, we employed the traditional Pearson correlation coefficient (hereafter, $\rho$).

## 2.6 Wavelet analysis

The Wavelet analysis is a very appealing alternative to the classic short time Fourier Transform for the geophysical time series analysis and periodicities examination. The Wavelet tool, in fact, allows discriminating not only the main frequencies in a non-stationary series, but also localizing them in time (Percival, 2008). This is a very useful feature for the analysis of climate data, whose variability is typically modulated by nonstationary processes. Because of such remarkable advantages, we have decided

to apply the Wavelet Transform (WT) to the clustered-averaged SCD and NDS time series involved in this study to identify their oscillation in the frequency domain. Among the main WT categories (Grinsted et al., 2004), we have chosen the Continuous Wavelet Transform (CWT), which is particularly suitable for the analysis of scale and time-dependant features of a time series. The CWT searches for a similarity between the investigated signals and a well-known mathematical function (the wavelet). The latter is applied several times with different scales to the considered time series and at different temporal locations. Similarly to other studies (e.g. Carey et al., 2013; Marcolini et al., 2017b), we have decided to apply the Morlet function as wavelet function. Additional details about the mathematical formulation of this function as well as about the wavelet analysis and the methods used to assess the statistical significance of the power spectrum can be found in Grinsted et al. (2004). It is important highlighting that CWT presents some deficiencies at the edges of the investigated time series. Therefore, it is useful to introduce a cone of influence, where the results are uncertain (Torrence and Compo, 1998). The CWTs of two signals can be combined to obtain the Cross Wavelet Transform (XWT), which offers the possibility to detected the areas, in the time-frequency domain, where the two CWTs share common features in terms of power and relative phase (for mathematical details, see Grinsted et al., 2004).

## 3 Results

### 3.1 Regionalization

The PCA of monthly SCD, NDS and HN time series allowed extracting the essential modes of spatial variability. A detailed description of the Principal Component Analysis (PCA) results for each of the three investigated variables is offered in Appendix B. According to the evidence provided by the scree plot, we have considered the first four PCs for SCD (which account for the 73% of the total variance), the first nine PCs for NDS (which explain the 70% of the total variance) and the first four for the HN (which account for 70% of the total variance). Such levels of explained variance seem to be acceptable, given the considerable number of stations and the complex nature of the snow variables. It is interesting highlighting that, in all three cases, the first PC explains most of the variance (61.2% for SCD, 51.8% for HN and 47.7% for NDS). From the analysis of PC scores, which can be regarded as standardized spatial patterns, it clearly emerges that the first PC reflects the altitude-related variability of the three snow indicators across the whole elevation range (see Figure B2, B4 and B7 in Appendix B). The other considered PCs explain a very limited portion of variance (generally ranging between 1 and 8%) and might be linked with several factors, such as the difference between eastern and western sides of the Apennines, the distance from the sea, the hours of direct sunlight, as well as with local patterns strongly conditioned by the orography.

The PCA scores from the selected PCs were employed as input of the k-means algorithm. After several tests and sensitivity analysis to assess cluster reproducibility and stability, four clusters were chosen for SCD and NDS and three for HN. The clustering of the available stations based on monthly SCD, NDS and HN data is presented in Fig. 4. From a visual inspection of this figure, as well as of Table 2, it is straightforward notice that the clusters are relatively well separated in altitude. Starting

from SCD (Fig. 4a), the identified groups are populated by a number of stations that is inversely proportional to the median elevation. More specifically, as revealed by Table 2, the cluster 1 comprises 58 stations and has a median altitude of 648 m ASL (min-max: 288-910 m ASL); the cluster 2 counts 26 stations having a median elevation of 815 m ASL (min-max: 550-954 m ASL); the cluster 3 includes 14 stations and has a median elevation of 1000 m ASL (min-max: 750-1157 m ASL), and, finally, cluster 4 includes 12 stations having an altitude greater or equal to 1000 m ASL (median value: 1290 m ASL). The altitudinal range of the stations is around 400 m for all clusters, except the cluster 1. The latter, in fact, includes not only low-elevation sites but also stations located on eastern and southern mountain slopes of the Central Apennines chain, which are located in topographic contexts unfavourable to the persistence of snow on the ground due to the high number of hours of direct sunlight.

The four groups emerged from the clustering of NDS data are shown in Fig. 4b. According to Table 2, most of the stations are nearly equally distributed among the first three clusters (31 belong to cluster 1, 34 to cluster 2 and 35 to cluster 3). The cluster 1 includes low-elevation sites (median altitude 569 m ASL, min-max: 288-850 m ASL) and stations located in the main valleys of Abruzzo region. The second cluster has a slight higher median elevation (700 m ASL, min-max: 520-910 m ASL) and brings together a relevant number of hilly sites of the Southern and Samnite Apennines, as well as several stations located on the eastern side of Majella mountains. The stations belonging to the third cluster span a significant altitudinal range (from 650 to 1375 m ASL) and a have a median elevation of 879 m ASL. The fourth cluster is very similar to the SCD cluster 4: it includes, in fact, only high-elevation sites having a median elevation of 1220 m ASL (min-max: 1000-1430 m ASL).

The clustering of HN data yielded three regions (Fig. 4c): the first one (cluster 1) includes 52 stations having a median elevation of 603 m ASL (min-max: 288-948 m ASL), the second one (cluster 2) comprises 45 sites having a median altitude of 829 m ASL (min-max: 625-1137 m ASL), whereas the cluster 3 includes 23 sites, all located in the Central Apennines (except one, Montevergine Observatory, which is situated in the Southern sector), having a median elevation of 1210 m ASL (min-max: 800-1750 m ASL). This classification is clearly conditioned not only by altitude but also by local climatic and topographic factors, which may generate large difference in snowfall amount among stations located at similar altitudes. Most of stations belonging to cluster 4 are located in the proximity of Majella and Marsicani mountains, where the orographic forcing and low-level wind convergence lines are able to strongly enhance the snow precipitation.

### 3.1 Climatology for 1971-2000 period

Starting from the clustering results, we have built up a reference climatology for different snow seasons: early season (from November to January), winter season (from December to February), late season (from February to April) and full season (from November to April). This seasonal partition is similar to that generally adopted in previous studies (e.g. Matiu et al., 2021), with some small adjustments depending on the climatic features of the study area and on the elevation range of the available stations. Fig. 5 presents the climatology 1971-2000 for SCD parameter. More specifically, for each of the four seasons previously introduced, the altitudinal distribution of the average SCD (expressed in number of days) is shown. Each point represents one station and is color-coded according to the membership cluster (note that we follow the same color-coding

adopted in Fig. 4). In addition, on each panel the cluster-averaged values and the associated spatial standard deviation are also reported. As it could be expected, there is a clear altitudinal gradient that grows as the elevation increases. In full season, the SCD average rises by ≈ 9 days from cluster 1 (which presents a climatological value of 12.9 (± 3.8) days) to cluster 2 (22.0 (± 3.0) days) and by ≈ 11 days from cluster 2 to cluster 3 (33.4 (± 4.6) days), whereas the difference among cluster 3 and cluster 4 (59.4 (± 12.1) days) is steeper (it is 26 days). The relationship between average SCD and elevation is efficiently modelled by a power fit, as testified by the coefficient of determination ($R^2$), which ranges between 0.72 (early season) to 0.78 (late season). Note that the power curve function obtained for a determined season is sketched on the corresponding panel as black solid line, assuming that the average snow cover duration is the dependent variable (*SCD*) and the elevation the independent one (*z*). It is interesting highlighting that the differences among clusters reduce in the early season, whereas they are larger in the winter season. In addition, it is worth noting that in cluster 4 the distribution becomes more dispersed, as testified by the standard deviation values. The spatial distribution of the average seasonal SCD over the study area is sketched in Fig. A1 in Appendix A. In all seasons, the highest average SCD value has been found in Camposto (1430 m ASL), which is located in a plateau east of the Laga mountains. In this site, the local topographic conditions are particularly favourable to the persistence of the snow on the ground.

The climatology for NDS is presented in Fig. 6. The average NDS values found for the available stations spread over the considered altitudinal range following, also in this case, a distribution well captured by a power law function ($R^2 = 0.78$ for all seasons). This behaviour suggests the existence of an altitudinal gradient, which, similarly to what has been discovered for SCD, rises with increasing of elevation. The cluster-averaged climatological NDS values (shown for each season on the corresponding panel) provide confirmation in this regards: in full season, the average NDS is 5.2 (± 0.7) days for cluster 1, 7.8 (± 1.3) days for cluster 2, 11.2 (± 1.9) days for cluster 3 and 20.0 (± 5.4) days for cluster 4. Above 800 m ASL, there is a clear increase of the spatial variability, which maximizes within cluster 4 in all seasons. The difference between clusters does not exhibit a relevant seasonal dependence, although in winter a slight increment of the gap in average NDS value between cluster 1 and 2 and between cluster 3 and 4 has been detected. It is worth pointing out that, in full and late seasons, the highest climatological NDS value has been observed in the Southern Apennines (Fig. A2), more precisely in Montevergine Observatory (1280 m ASL). This result is only apparently surprising. As briefly discussed in Section 2.1, the Southern Apennines massifs included in the investigated area are well exposed to different air masses (i.e. to both continental air masses coming from Balkan regions and to maritime air masses coming from the Atlantic), so they receive larger precipitation amounts than the Central sectors. According to the results of our study, this aspect also reflects on the frequency of occurrence of snowfall events, which, on Partenio massif, is larger than many mountainous sites of the Central Apennines, located at the same altitude or slightly above, as demonstrated by Fig. A2.

The HN climatology is presented in Fig. 7. As for SCD and NDS, the analysed region exhibits a pronounced variability, mainly driven by the elevation, with a minimum, in the full season, of 24 cm in Lanciano (288 m ASL) and a maximum of 335 cm in Monte Terminillo station (1750 m ASL), in the Reatini Mountains. According to the scatter diagrams of Fig. 7, there is a clear separation between the three groups identified by the cluster analysis. In the full season, the climatological HN value is 44.5

(± 11.7) cm for cluster 1, 85.0 (± 18.5) cm for cluster 2 and 204.1 (± 51.9) cm for cluster 3. The latter shows a remarkable variability, which is related not only to the altitude but also to the incidence of orographic effects. In this respect, in the Abruzzo region there are several stations below 1000 m ASL that received, in the considered reference period, relevant snowfall amounts, comparable to sites belonging to higher altitudinal bands. This is the case of Nerito (800 m ASL), in which the full season climatological HN value is 148 cm, Rosello (890 m ASL), in which the average HN value is 167 cm and Sant'Euefemia

a Majella (870 m ASL), the most impressive case, where the average HN value is 268 cm (Fig. A3). In the late and winter seasons, the snowfall amounts increase by ≈ 80% from cluster 1 to cluster 2 and by ≈ 124% from cluster 2 to cluster 3. In late season, we have detected a steeper altitudinal gradient, instead. The cluster 2 receives, on average, more than twice (+109.7%) the total HN value observed for cluster 1, whereas in cluster 3 the snowfall amounts grow by 150% with respect to cluster 2.

### 3.3 Trend analysis

The linear trend analysis has been applied to both individual SCD and NDS time series as well as to cluster-averaged time series. The latter have been obtained, for each of the investigated seasons, as the arithmetic mean of the values of stations belonging to a determined cluster.

Starting from SCD, Fig. 8 shows the cluster 1 (green solid line) and cluster 4 (blue solid line) time series for 1951-2001 period. Note that we decided not to plot the behaviour over time of cluster 2 and 3 only for ease of presentation. According to Section

3.1, cluster 1 and cluster 4 include stations belonging to different altitudinal bands (the median elevation is 648 and 1290 m ASL, respectively) and, therefore, they reflect two well-separated nivometric regimes. From a simple visual inspection of this figure, it clearly emerges that the investigated signals exhibit a negative tendency (see the black solid line). More specifically, the trend magnitude (expressed in number of days/10 years) is more pronounced in full season (−3.4 [−7.3 to 1.7] days/10 years for cluster 4 and −1.1 [−2.6 to 0.2] days/10 years for cluster 1) and in winter season (−3.2 [−6.0 to 0.0] days/10 years

for cluster 4 and −1.1 [−2.5 to 0.2] days/10 years for cluster 1). Note that the values indicated in square brackets indicate the 95% confidence interval for linear trend. In the other seasons (early and late), the cluster 1 shows a negligible trend, whereas for cluster 4 a decreasing rate of −1.1 [−3.3 to 1.0] days/10 years and −1.8 [−4.5 to 0.6] days/10 years has been detected, respectively. As revealed by Table 3, the trends are statistically significant only in full and winter seasons. More specifically, in winter season a negative tendency significant at 90% confidence level has been found for clusters 1, 2 and 4, whereas for

cluster 3 the linear trend is significant at 95% confidence level. In full season, robust tendencies have been discovered only for cluster 1 and 4. According to Table 3, in all seasons the trend magnitude gradually increases from cluster 1 to cluster 4, so, in other words, it rises with increasing altitude, especially in full and winter seasons. In addition, Table 3 shows, for each cluster, the percentage of positive, negative and no trends (i.e. the subset of tendencies ranging between −0.4 and 0.4 days/10 years), and the portion of trends significant at 95% confidence level. As can be expected, most of stations exhibit a negative tendency.

In particular, for clusters 2, 3 and 4, the percentage of negative trends is above 60% in all seasons and is greater than 90% for clusters 3 and 4 in winter and for cluster 4 in full season. For cluster 1, there is a clear prevalence of negative trends only in full and winter periods, whereas in early and late seasons most of the stations belonging to such cluster present a negligible

trend. Moreover, the fraction of significant and negative tendencies has an altitudinal dependency that is more linear and evident in full and winter seasons. In the latter, half of the stations pertaining to cluster 3 and 4 exhibit a negative trend
significant at 95% confidence level. Another distinguishable behaviour of the signals sketched in Fig. 8 is the strong interannual variability, which is particularly pronounced in cluster 4. This is a distinct feature of the precipitation records collected in mid-latitudes that, with regard to Apennine area, has been just emphasized in previous works (Capozzi et al., 2022). Focusing on full season, in cluster 4 the highest SCD values have been observed in 1962/63 season (105.3 days), 1952/53 (98.2 days) and 1980/81 (95.8 days), whereas the lowest SCD values have been observed in 1989/90 season (10.6 days), 2000/01 (22.0 days)
and 1963/64 (26.0 days). The season 1962/63 was notable in terms of SCD also for cluster 1 (43.0 days), although the highest value for this group has been observed in 1955/56 (45.6 days), one of the coldest and snowiest seasons of the 20th century (e.g. D'Errico et al., 2022). In cluster 1, the lowest values have been recorded in 1989/90 (0.9 days), 1960/61 (2.6 days) and in 1963/64 season (2.7 days). Superimposed to the decreasing trend, the Apennine's SCD also shows considerable decadal variations, well captured by the 10-years locally weighted scatterplot smoothing (lowess). The latter, marked as red line in Fig.
8, is a robust non-parametric regression technique proposed by Cleveland (1979). By inspecting the behaviour of lowess fit, it is possible to detect four periods characterized by extensive duration of snow cover, the early 1950s, the 1960-1970 period, the late 1970s-early 1980s and the early 1990s, and four time segments with reduced SCD, the late 1950s, the mid-1970s, the late 1980s and the late 1990s. The decadal oscillation of SCD seems to be a common features of all considered seasons and is more relevant in cluster 4 time series.

The NDS time series exhibit a behaviour very similar to that described for SCD, as demonstrated by Fig. 9. For both clusters 1 and 4, we have detected a gradual decrease of the frequency of occurrence of snowfall events, again particularly pronounced in full and winter seasons. In the former, the linear trend analysis revealed a negative tendency of −1.7 [−3.0 to −0.5] days/10 years for cluster 4 and of −0.8 [−1.3 to −0.1] days/10 years for cluster 1, whereas in the latter the tendency magnitude is −1.6 [−2.5 to −0.6] days/10 years for cluster 4 and −0.6 [−1.2 to −0.1] days/10 years for cluster 1. Such trends are statistically
significant at 95% confidence level, as indicated by Table 4. Strong negative tendencies have been also found, in full and winter seasons, for the other cluster-averaged time series (except for cluster 2, whose trend is not statistically significant in full season). As for SCD, in late season cluster-averaged trends are negligible and range between −0.1 [−0.7 to 0.4] days/10 years (cluster 2) and −0.6 [−1.6 to 0.2] days/10 years (cluster 4). In the early season, negative trends significant at 90% confidence level have been detected for cluster 3 and 4 (trend magnitude is −0.6 [−1.2 to −0.1] days/10 years and −0.9 [−1.6 to −0.3]
days/10 years, respectively). Table 4 shows that long-term tendencies taking a direction towards NDS reduction are prevalent in all seasons for cluster 3 and 4, with percentage up to 97-100% in winter season, whereas this result is not valid for cluster 1 and 2, in which the no trend fraction predominates in the early and late seasons. The percentage of stations having a trend significant at 95% confidence level found for clusters 3 and 4 as well as the trend magnitude are substantially larger than that discovered for clusters 1 and 2. Such results suggest the existence of a relationship between long-term tendency and elevation
in terms of statistical significance and magnitude. The decadal trend emerged from the analysis of SCD time series is also evident in NDS signals, as highlighted by the lowess fit in Fig. 9. Similarly to what has been observed for SCD, this decadal

behaviour is more pronounced in cluster 4. The NDS signals also exhibited a very strong interannual variability, especially in the 1950-1960s. In cluster 4, the frequency of occurrence of snowfall was particularly higher in 1962/63 full season (40.1 days), in 1955/56 (36.8 days) and in 1969/70 (33.0 days). It is interesting highlighting that the second and the third snow
seasons more lacking in snow events occurred in the 1950-1960s: the 1963/64 (7.7 days) and the 1958/59 (9.7 days). The lowest NDS value has been detected in the 1989/90 season (5.1 days). The latter has been the weakest season in terms of snowfall occurrence also for cluster 1 (0.3 days). Very low values (1.6 and 1.4 days, respectively) have also been observed in 1963/64 and in 1960/61, respectively. The 1962/63 (21.5 days), the 1955/56 (18.1 days) and the 1953/54 (14.5 days) were the three richest seasons in terms of snow episodes for cluster 1, instead.

Linear trends for individual stations as function of elevation at seasonal time scales are sketched in Fig. 10 (SCD) and in Fig. 11 (NDS). Such diagrams provide more compelling evidence about the relationship between trend magnitude and altitude. As a general result, a moderate correlation between the two variables has been found for both snow indicators. More specifically, for SCD, the correlation is stronger in late and winter seasons ($\rho = |0.63|$ and $|0.62|$, respectively), whereas it is slightly weaker in late season ($\rho = |0.43|$). For NDS, different results emerged in terms of seasonal variability of the correlation. The latter
maximizes in early season ($\rho = |0.59|$), whereas it is lower in late season ($\rho = |0.46|$). A visual comparison between Fig. 10 and 11 also reveals that SCD trends are characterized by a larger variability than NDS, especially at altitudes greater than 1000 m ASL. It is worth notice that in full season the maximum negative trend ($-7.6$ [$-11.8$ to $-2.4$] days/10 years) was found for Capracotta (1400 m ASL), a station belonging to cluster 4. The maximum positive tendency (1.4 [$-3.7$ to 6.0] days/10 years) has been detected for Pietracamela (1000 m ASL), a station that is part of the same cluster. This result well synthetizes the
strong variability and uncertainty that affect the linear trend estimation at high elevation ranges, which can be interpreted as a consequence of the strong year-by-year fluctuations as well as of the local orographic effects incidence.

**3.4 Connections with large-scale atmospheric patterns: preliminary evaluations**

A natural evolution of our study is a further investigation aimed to identify the main drivers controlling the SCD and NDS
variability in the considered Apennines region. Recently, several research activities (e.g. Hammond et al., 2018; Annella et al., 2023; Bertoldi et al., 2023) dealt with this topic, taking into account both "local" variables, such as temperature and precipitation, as well as "global" drivers such as the large circulation patterns. According to the general results achieved for the Alpine region, the role of temperature and precipitation in controlling the snow presence is strongly modulated by the elevation. Generally, at low elevations most of the snow variability is explained by temperature, whereas at high elevation the
precipitation has a greater relative importance (e.g. Bertoldi et al., 2023). However, in a very recent study that documented the exceptional snow-drought conditions that affected the Italian Alps during the winter 2021/22, Colombo et al. (2023) discovered an increasing relevant role of warming air temperature in driving the snow-drought events in the whole investigated elevation range (864-2200 m ASL). For the Apennines, Annella et al. (2023) found that the reduction in SCD observed in Montevergine Observatory could be mainly attributed to the increasing trend in temperature, which was statistically significant in winter and

late seasons in the considered time interval (1931-2008). In light of such results, it may be speculated that the decline in SCD and NDS discovered in our study has been mainly driven by the rising temperature tendency occurred in the second half of XX century. Currently, we are not able to demonstrate this assumption with a deep investigation. Most of the historical temperature and precipitation records collected in the study area, in fact, are not accessible into a ready-to-use digitized format. Therefore, a complete attribution analysis is left for future analysis. However, here we discuss some preliminary linkages

between the decadal trend observed in our study for SCD and NDS signals and the large-scale circulation patterns. More specifically, our analysis starts from the results of Annella et al. (2023). From this study, it emerged that short-time and decadal fluctuations of SCD in Montevergine Observatory are strongly modulated by two teleconnections patterns, the Arctic Oscillation (AO) and the Eastern Mediterranean Pattern (EMP). The first one is a fundamental mode of the Northern Hemisphere climate variability as it describes simultaneous shift in several features of the polar vortex (air temperature, air

pressure and the location and strength of the jet stream). The EMP has been introduced by Hatzaki et al. (2007) and is referred as the difference in 500-hPa geopotential height anomaly between the Eastern Atlantic and the Eastern Mediterranean. Using the Wavelet tool, we search for relationship between these two large-scale patterns and SCD and NDS signals. Note that the corresponding index for AO (the AO index, hereafter AOI) has been retrieved from Climate Prediction Centre of the NOAA's National Weather Service (Climate Prediction Centre, 2024), whereas the EMP index (hereafter, EMPI) was reconstructed in

Capozzi et al. (2022) using the version 3 of the Twentieth Century Reanalysis dataset (Allen et al., 2011), following the method described in Hatzaki et al. (2009). The XWT between AOI and SCD cluster-averaged time series did not reveal noticeable results, except for winter season, in which a significant common power area on 2-year band has been detected between 1960 and 1965. In this region of the time-frequency spectrum, the AOI and SCD are in anti-phase relationship. More interesting evidence came from the analysis of the relationship between EMP and SCD. Fig. 12 presents the XWT between EMPI and

SCD cluster 4 time series for all four considered seasons. We have chosen the cluster 4 SCD time series as reference for this analysis because its behaviour in the time-frequency spectrum can be regarded as highly representative of the one observed for other three clusters. This aspect has been confirmed by the CWT, which depicts a coherent picture among the clusters, characterized by a significant peak in the ≈12-14 year band from 1960s to 1990s and by two high-frequency (≈2 year) oscillations, one located between 1951 and 1955 and the other one between 1960 and 1965. The only exception is the cluster

1, in which the decadal oscillations are not statistically relevant. Starting from full season (Fig. 12a), it is easy to detect a significant common power in the ≈12-16 year band from 1955 to 1985 and between 1960 and 1965 in the ≈0-2 year band. A close connection between EMPI and SCD can be also found, on ≈12-14 year band, in the early season (Fig. 12b) from 1965 to 1985, whereas in winter (Fig. 12c) the common power area on decadal scale is restricted between early 1960s and early 1970s and falls entirely within the cone on influence. A very strong and significant connection between the investigated signals

has been found in late season (Fig. 12d) on ≈12-16 year band. The common power area, in this case, extends from late 1950s to mid-1990s, so it embraces almost the entire analysed period. Some linkages in the high-frequency region has been also detected between 1955 and 1965 (≈5 year), in 1985-1990 (≈0-2 year band) in early season, between late 1970s and early 1980s (≈0-2 year band) in winter season, and, finally, in the 1960-1965 and in the early 1970s (≈0-2 year in both cases) in

late season. The right pointing black arrows in the significant power areas indicate that EMPI and SCD clearly swing in phase on decadal time scale, adding further evidence about the close time-frequency connection among the signals. The XWT between EMPI and NDS cluster 4 time series draw a similar picture, as testified by Fig. 13. In this case, there is a close and significant connection on decadal scale only in full season (Fig. 13a) and in late season (Fig. 13d). The common power areas are localized in the same spectrum regions mentioned for SCD. In the ≈0-2 year band, relevant connections between EMPI and NDS have been found in 1961-1965 period in all seasons, between late 1970s and early 1980s in winter (Fig. 13c) and in the early 1970s in late season. In early season (Fig. 13b), a common power area also appear in the ≈5 year band between mid-1950s and mid-1960s. The two signals are generally in a close in-phase relationship, except for some significant areas in the high-frequency region, in which there is a slight lag between the two signals.

The direct in-phase connection existing between the investigated snow indicators and the EMPI is in line with what it should be reasonably expect about the influence of EMP on snow variability in the Central and Southern Apennines. The positive phase of EMP pattern, in fact, is associated with positive 500-hPa geopotential anomalies over northern Atlantic and with negative ones over Central and Eastern Mediterranean basins (Hatzaki et al., 2007). This synoptic pattern generally drives Arctic or Polar cold continental air masses towards Italy and, therefore, is clearly favourable to snow occurrence and persistence on the ground in the Apennines, as previously demonstrated in Capozzi et al. (2022) and in Annella et al. (2023). The negative phase of EMP depicts a very different configuration, which generally brings mild weather conditions in the considered area.

## 4 Discussions

In this study, thanks to the rescue of a large amount of manually snow observations collected by NHMS during the 1951-2001, it was possible to build-up a new, updated and solid reference climatology for an area, the Apennine regions, in which the information about past nivometric regime are scarce and very fragmented. For all considered snow indicators and for all investigated seasons, we found a relevant altitudinal gradient that grows as the elevation increases. This result exhibited a seasonal dependence for SCD and HN. More specifically, in the first case the altitudinal gradients are steeper in winter and reduce in the early season, whereas for HN the altitudinal gradient is more pronounced in late season. The clusters including the stations above 1000 m ASL showed a strong spatial variability, mainly related to the orographic effects. In this respect, our results are in accordance with some previous works (e.g. Blanchet et al., 2009; Bertoldi et al., 2023), related to the Alpine region.

Our analysis has revealed that in the considered Apenninic region the SCD and NDS parameters exhibited a similar behaviour in terms of long-term and decadal trend. For both variables, a decreasing tendency has been detected in the 1951-2001 period. The observed variations are strongly connected with the season (i.e. they are more relevant in winter) and show a marked dependence from the elevation. It is obviously not straightforward to contextualize such results in the available literature,

because linear trends magnitude and their statistical significance are strongly dependent from the analysed time window. Focusing on papers that considered periods having a good overlap with the present work, it clearly emerges that our study confirms the local and general tendencies observed for SCD and NDS in the Mediterranean area. Regarding SCD, the declining tendency highlighted in this study is in agreement with the local trends found for Apennines (e.g. Petriccione and Bricca, 2019; Annella et al., 2023), as well as with the outcomes presented for several Alpine regions (e.g. Klein et al., 2016; Marcolini et al., 2017b; Marke et al., 2018; Matiu et al., 2021; Bertoldi et al., 2023). Another common point between our results and previous works lies in the elevation dependence of SCD trends. More specifically, this aspect has been discussed in Marcolini et al. (2017b), which analysed SCD and HS time series collected in the Adige catchment (North-East of Italy) from 1980 to 2009. They found a reduction in both variables at low and high altitude sites, although a difference emerged between the behaviour of stations located above and below 1650 m ASL. In sites located below this altitude threshold, the decline in both SCD and HS was larger. This work concludes that areas under 1650 m ASL are more sensitive to climate variability and to temperature increase than high elevation regions. Matiu et al. (2021) have reported a similar and more generalized result for the Alpine region: they found a decreasing trend in SCD below 2000 m ASL, while above no remarkable variations have been detected, at least in the period from November to May. The elevation dependence of snow trends has been also highlighted in a very recent work (Bertoldi et al., 2023) focused on the northeastern Italian Alps. Although this study is focused on HN indicator, it reported evidences comparable, at least in part, with SCD-based studies. On monthly basis, negative trends were found in the lowest elevation range (0-1000 m ASL), some positive trends from January to March above 2000 m ASL, while the intermediate elevation band (1000-2000 m ASL) showed a strong variability with no robust tendencies. Averaged seasonal trends are negative for all elevation ranges, instead: in absolute terms, the maximum negative trend was found at intermediate levels.

Unfortunately, the stations available in our study cover a limited altitudinal band (the only station situated above 2000 m ASL, Campo Imperatore, has a very limited data availability), so we are not able to reconstruct the behaviour of SCD and NDS trends over the elevation range considered in previous studies for Alps and to identify a critical altitudinal band that separates different regimes characterized by opposite trend direction. Overall, the results of our study show that low and intermediate Apenninic levels (288-1430 m ASL) are experiencing a decline in SCD that is similar to what generally observed in the Alps. However, in our case, most of the significant trends have been found in the core of the snow season (i.e. in the winter), whereas in the Alps the percentage of negative trends was substantially higher in spring months (e.g. Matiu et al., 2021).

The NDS is a less-studied variable than other snow indicators, such as SCD, HN and HS. Our results confirm some evidence found in Switzerland by Marty (2008). From this study, in fact, a long-term downward tendency in snow days emerged for the 1948-2007 period. The decreasing NDS signal is stronger in the low altitude zone, whereas the higher stations showed a marked variability. Terzago et al. (2010) also found a decrease in NDS for the Piedmont region (1971-2009 period). In a subsequent work, related to a more extended period (1926-2010), Terzago et al. (2013) discovered a decline of the fraction of precipitation falling as snow for the Western Alps; in line with our study, the dropping in snow events was found to be more relevant in winter season.

It is interesting pointing out that the linkages between EMP and snow indicators described in Section 3.4 has not been reported in any previous work related to the Alpine region. For this area, most of the available studies searched for connections with other large-scale circulation patterns, such as the North Atlantic Oscillation (NAO), the AO and the Mediterranean Oscillation. The results are generally ambiguous and strongly dependent on the considered region and time interval (e.g. Durand et al., 2009, Kim et al., 2013; Marcolini et al., 2017b; Bertoldi et al., 2023). For the 1930-2020 period, Colombo et al. (2022) found that NAO, Winter NAO, Atlantic Multidecadal Oscillation and AO were anticorrelated with the Standardized Snow Water Equivalent Index, during different phases of snow season.

Regarding the AO, from the preliminary results presented in this study, based on the Cross Wavelet Transform, it emerged that it exerts a less relevant influence than EMP on the nivometric regime of the investigated Apennine region. However, we feel that additional analyses are necessary to better assess the relationships between this important atmospheric mode and the snow variability in the study area. Two previous studies dedicated to the Apennines (Capozzi et al., 2022; Annella et al., 2023), in fact, found that the recovery in some snow indicators observed after 2000 is closely linked to AO trend. It is possible to assume that non-negligible difference might exist between western and eastern sectors of the Apennines (the first ones might be more "sensitive" to the AO variability).

As stated in the Section 1, ground observations are crucial for assessment of long-term snow variability and trends, especially in mountainous areas. However, it is necessary bear in mind that manual snow measurements have several limitations, mainly related to observer's errors in reading and recording the measurement and in poor siting (WMO, 2008). A possible source of uncertainty that affect the dataset rescued in this study, and consequently the results just discussed, is related to the incorrect counting of snow days in a determined month (i.e. NDS parameter). As an example, the observer might have considered as snowy, a day in which snowfall occurred without leaving any trace on the ground. In addition, in many contexts the HN measurements can be very doubtful due to the environmental conditions, such as turbulence and/or strong winds, which may generate snow drifts and spatial inhomogeneity in snow depth.

**5 Conclusions**

According to the Intergovernmental Panel on Climate Change (IPCC) report on high mountain (Hock et al., 2019), a general decrease in snow cover duration, glaciers and permafrost due to climate change has occurred in last decades. The strong loss in mountain cryosphere is likely to have relevant repercussions for global population who rely on the water stored in mountain snow and ice for their water supply. Despite the serious impacts of mountain cryosphere loss, for several reasons many mountain areas remain under-researched. In the Mediterranean, an example in this sense is represented by the Apennine region. A considerable lack, in fact, exists in the knowledge of the past snow variability for this area, although it has a good heritage of past in situ observations. This study has provided a contribution to bridge this gap, through the rescue and the analysis of the snow precipitation measures manually collected between 1951 and 2001 by the Italian National Hydrological and

Mareographic Service in an area including a large part of the Central and Southern Apennines. After being subjected to quality check and homogenization procedures, the rescued dataset, consisting of monthly data of snow cover duration (SCD), number of days with snow (NDS) and height of new snow (HN), has been primarily analysed to retrieve a reference climatology (1971-2000 period). To pursue this aim, using a methodology based on Principal Component Analysis and k-means clustering, for each snow indicator we have grouped the available stations in different clusters. This classification has been mainly driven by the altitude and, secondly, by other factors controlling the spatial variability of snow precipitation, such as the distance from sea, the site exposure, the hours of direct sunlight and local orographic features. The presented snow climatology greatly enhances and expands the existing historical database of several key snow-related variables, SCD, NDS and HN, for which continuous and high-quality measures are difficult to find. In addition, it constitutes an added value for researches focused on the comprehension of climate dynamics in mountainous area as well as on future changes in snow precipitation in the Mediterranean Region and provides useful information for a wide range of application fields, concerning also the socio-economic impacts of snow precipitation.

Furthermore, using familiar statistical methodologies (the Sen's slope estimator and the Mann-Kendall test), we have identified the linear trend for SCD and NDS time series (the HN series have not considered for this analysis due to the limited length of the available records). Both variables exhibited a negative tendency for the 1951-2001 period. We found that SCD and NDS trends are strongly dependent on altitude, in terms of magnitude and level of confidence. More specifically, the signal of decrease in length of snow cover on the ground and in the frequency of occurrence of snowfall gradually grows with the altitude and is generally very strong for stations located above 1000 m ASL. Considering the entire snow season (i.e. the full season, from November to April), SCD trends statistically significant at 90% confidence interval have been discovered for cluster 1 (−1.1 [−2.6 to 0.2] days/10 years) and for cluster 4 (−3.4 [−7.3 to 1.7] days/10 years). For NDS, trends significant at 95% confidence interval have been detected for cluster 1, 3 and 4 (−0.8 [−1.3 to −0.1] days/10 years, −1.2 [−2.2 to −0.2] days/10 years and −1.7 [−3.0 to −0.5] days/10 years, respectively). At seasonal scale, the larger fraction of negative and significant trends has been found in winter season for both variables. In early and late seasons, the aliquot of significant tendencies strongly reduces, especially in clusters 1 and 2. In addition, we found that SCD trends exhibit a more pronounced variability and uncertainty than NDS, especially in cluster 4 (which includes, for both snow indicators, only stations above 1000 m ASL). In the considered Apenninic area, the SCD and NDS variables also show fluctuations at decadal scale as well as a remarkable interannual variability, in accordance with the findings of previous studies (e.g. Annella et al., 2023). The decadal behaviour gradually emerges with increasing altitude and is particularly relevant in cluster 4.

Despite some uncertainties and source of errors, which have been briefly discussed in the previous section, this study can be considered the first wide-Apenninic assessment of snow climatology and long-term trend based on in situ observations. The information provided by this work add a contribution to the reconstruction of historical snow variability in the mountainous areas and pave the way for many future research activities. In this respect, future studies will be primarily devoted to deeply understand the physical mechanisms that control the evolution over time of the investigated snow variables. Regarding this aspect, we provided some preliminary results by means of a Wavelet Analysis. More specifically, from the Cross Wavelet

Transform of Eastern Mediterranean Pattern Index (EMPI) and cluster 4 SCD and NDS time series, it emerged a significant common power in the ≈ 10-16 year band. The two signals are phase locked, so we can conclude that on decadal scale the SCD and NDS behaviour in the investigated Apennines area mirrors the evolution of EMP. Future analyses should be oriented to better assess the influence of other teleconnection patterns (in particular the Arctic Oscillation) on the observed interannual variability.

In addition, other efforts may be addressed to (i) extend back and forward the investigated period, in order to further increase the robustness of trend analysis and to contextualize the observed SCD and NDS tendencies in a broader time horizon, and (ii) to replicate this study for the northern Apennine sector and for the remaining sector of the Southern Apennine through the rescue of nivometric stations belonging to the remaining Italian National Hydrographic and Mareographic Service compartments.

# Appendix A

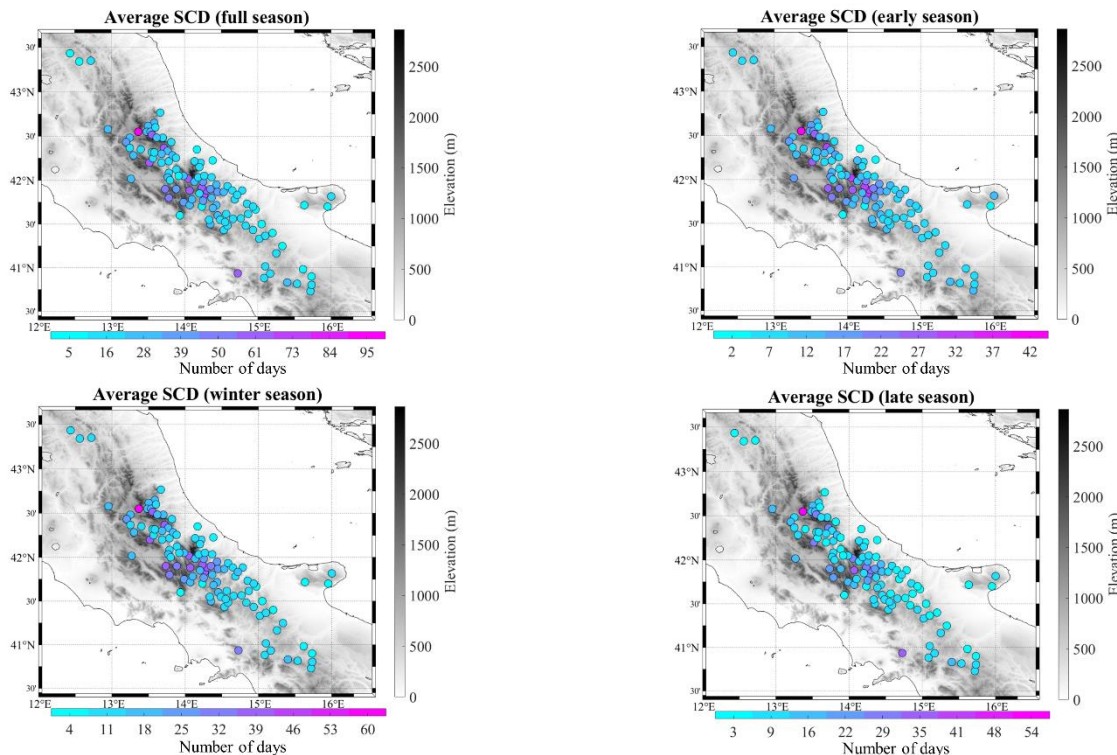

**Figure A1: Spatial distribution of the average seasonal snow cover duration (SCD) over the study area in the period 1971-2000. Each point represents one station and the corresponding climatological value.**

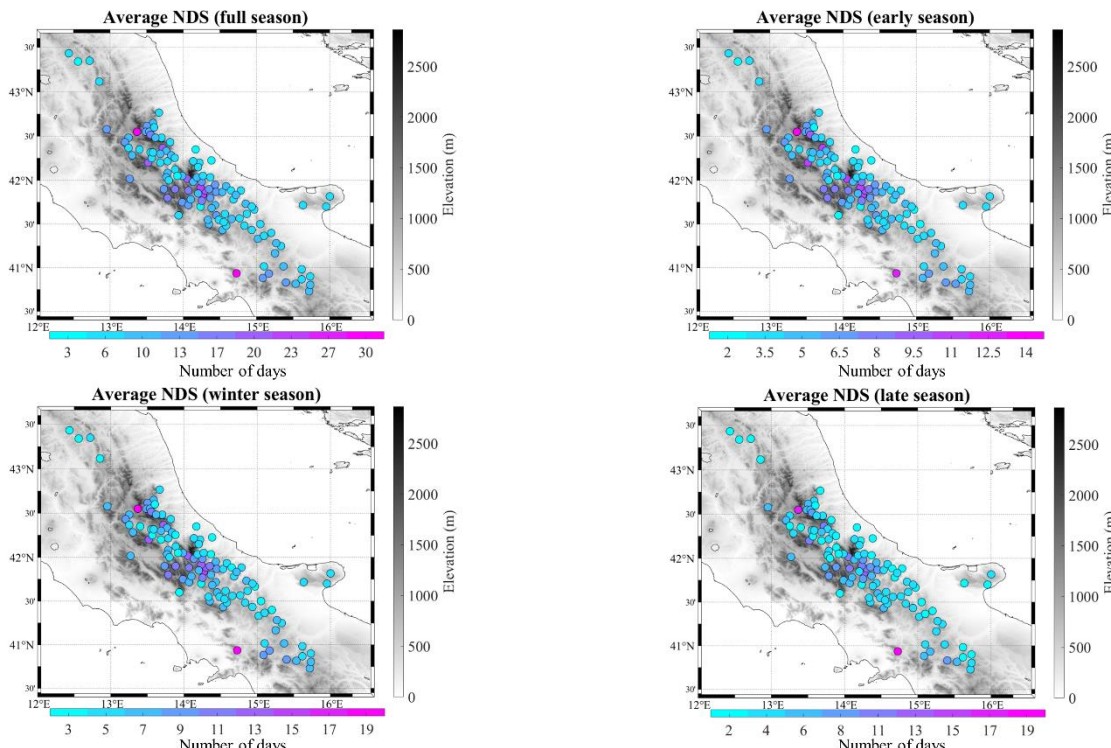

770

**Figure A2: Spatial distribution of the average seasonal number of days with snow (NDS) over the study area in the period 1971-2000. Each point represents one station and the corresponding climatological value.**

775

780

785

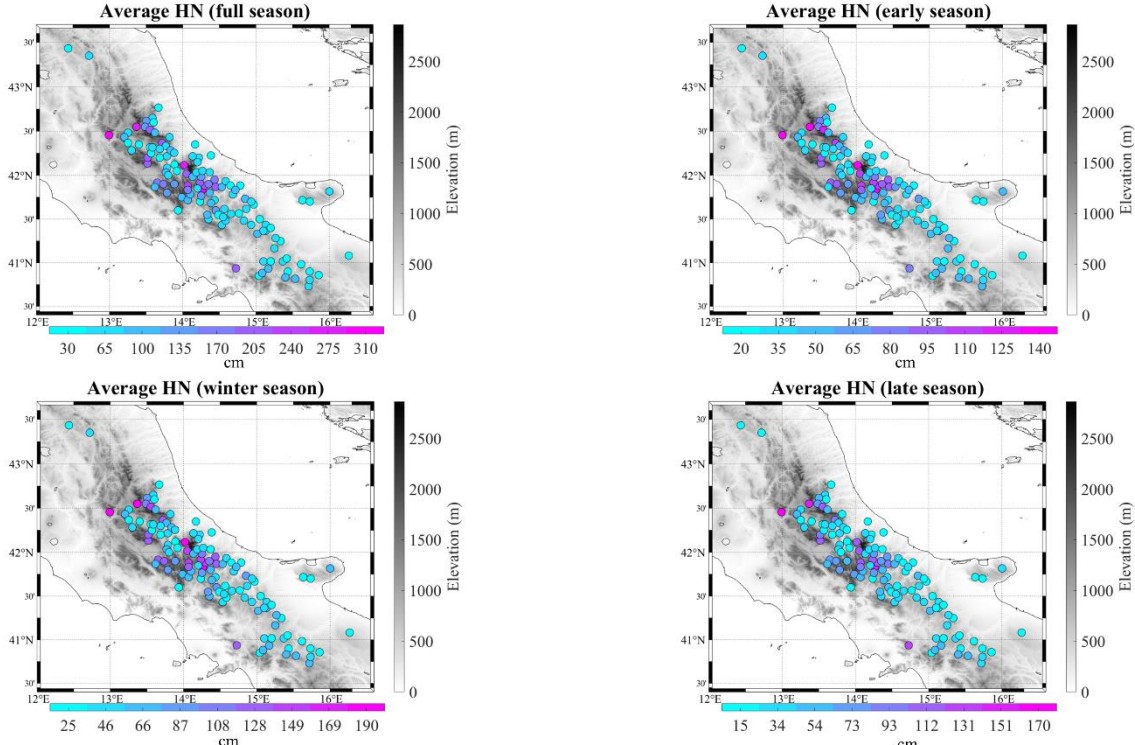

**Figure A3: Spatial distribution of the average seasonal total height of new snow (HN) over the study area in the period 1971-2000. Each point represents one station and the corresponding climatological value.**

**Appendix B**

In this Appendix, we provide a detailed description of the Principal Component Analysis (PCA) results for each of the three investigated variables: snow cover duration (SCD), number of days with snowfall (NDS) and height of new snow (HN).

Starting from SCD, the first PC (Fig. B1a), which represents the 61% of the total variance, reflects the altitude-related variability across the whole elevation range. Areas with positive scores coincide with some of the main mountain ridges of the considered region (Gran Sasso, Marsicani, Majella and Partenio). Negative scores mark low-elevation areas as well as the eastern and southern mountain slopes of the Central Apennine chain, where the local topographic features are not favourable to the persistence of snow on the ground. More compelling evidence about the relationship between PC1 and elevation is provided by Fig. B2, in which the PC1 scores are plotted against the altitude. A solid positive correlation was found (the linear correlation coefficient, $\rho$, is equal to 0.87).

The PC2 (Fig. B1b) separates the Central Apennine sector (Abruzzo and Molise regions) from the Southern area. In the first one, the scores are generally positive, whereas in the second one they are slightly negative. The high positive scores found in

several sectors of Abruzzo and Molise (mainly in the Gran Sasso and Marsicani areas) indicate relevant positive SCD anomalies.

PC3 spatial pattern (Fig. B1c) is characterized by a clear west-east gradient. More specifically, positive scores have been found in the Majella area, in the eastern side of Marsicani mountains and in the eastern side of Molise and Southern Apennine. In the western sector of Abruzzo region, negative scores prevail, instead. This pattern might reflect specific large-scale atmospheric weather regimes, associated with the incoming, over the study region, of cold continental air masses from the Balkan Peninsula. Such atmospheric scenario promotes conditions favourable to the occurrence and persistence of snow on the ground over eastern slopes of Apennine.

In the PC4 spatial pattern (Fig. B1d), the scores are generally around 0.0, except for the northern side of Abruzzo (Gran Sasso mountains). This pattern might reflect specific atmospheric conditions that enhance the snow duration on the ground only in high-elevation sites of the northern Abruzzo region.

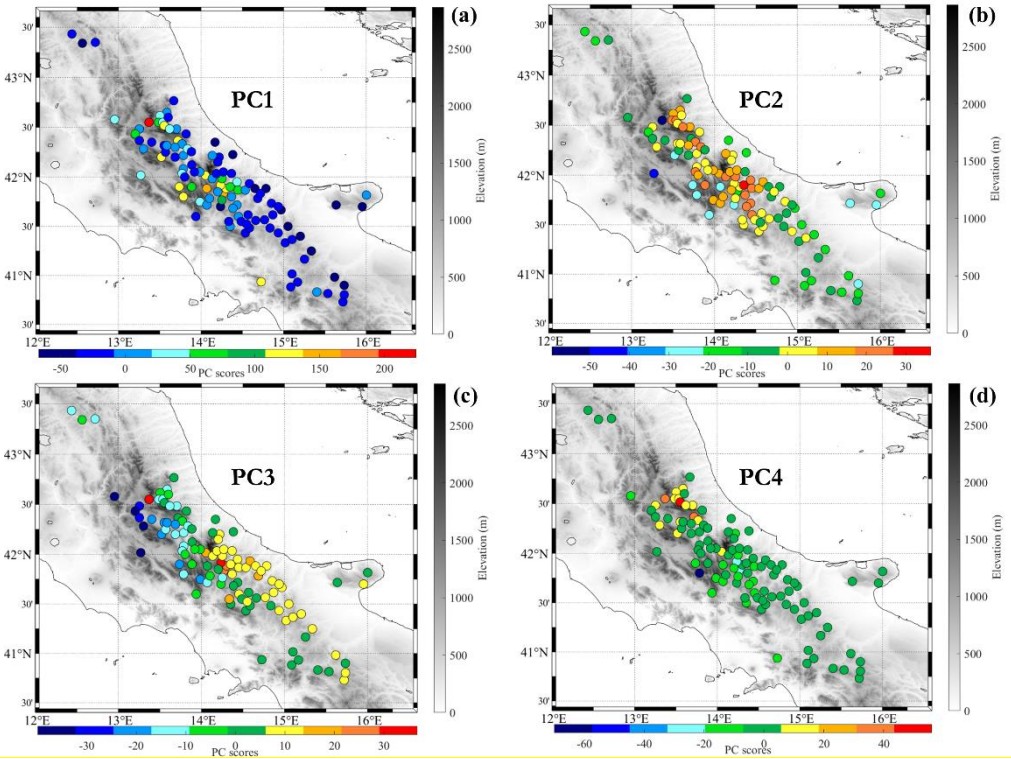

**Fig. B1. Spatial patterns of the first four modes resulting from the Principal Component Analysis applied to monthly SCD data.**

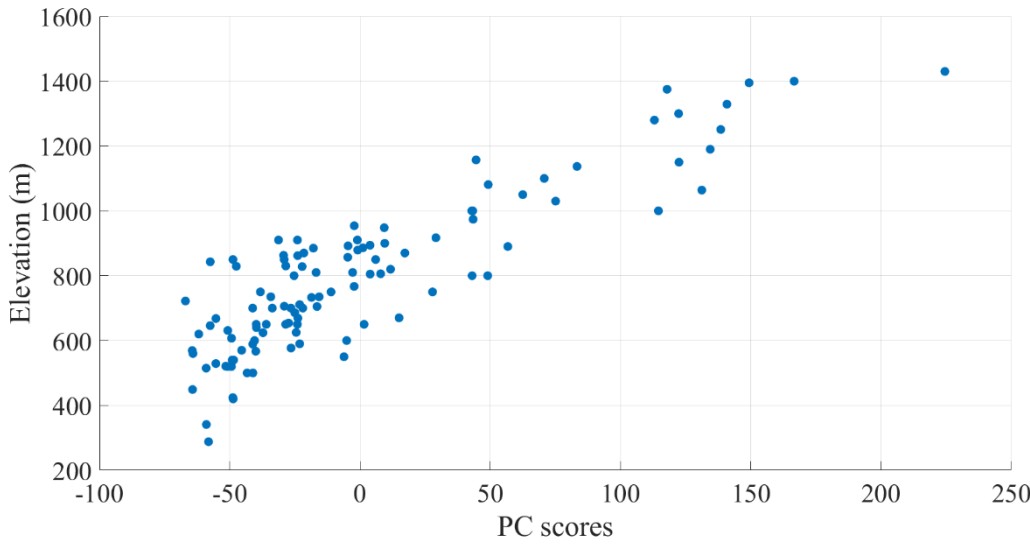


**Fig. B2. First principal component (PC1) scores resulting from PCA applied to monthly SCD data as function of the elevation (in m). Each point represents one station.**

For NDS variable, we have selected the first nine PCs, which capture the 70% of the total variance. According to Fig. B3a, the first PC represents a scenario in which the spatial distribution of the considered parameter is strictly related to the elevation.

In this sense, additional evidence comes from Fig. B4, which clearly demonstrates the strong relationship between PC1 scores and elevation ($\rho = 0.87$).

In the PC2 spatial pattern (Fig. B3b), there is a relevant gradient in terms of PC scores in the Abruzzo region. More specifically, the scores gradually switch from negative to positive values moving eastward. Areas with positive scores match with Majella, Marsicani, Matese and with Southern Apennine reliefs (Partenio, Picentini and Lucania mountains). It may hypothesize that

behind this NDS spatial pattern there is a synoptic scale atmospheric circulation scheme like that described for PC3 of SCD variable, i.e. a configuration associated with the incoming, over the Italian Peninsula, of cold air masses from Balkan region.

In the PC3 spatial pattern (Fig. B3c), the scores are negative over a large part of the study area. Positive values are restricted to the Campania Apennine (Partenio and Picentini mountains). Therefore, this spatial pattern might represent meteorological scenarios in which the snowfall events mainly affect the meridional sector of the considered area.

The PC4 (Fig. B3d) exhibits a spatial structure close to PC2. However, in this case the zonal gradient is not limited to the Abruzzo region, but it is extended to the whole area. As for PC2, scores gradually increase from west to east, so the largest values have been found on the eastern slopes of Apennines and over the Gargano area.

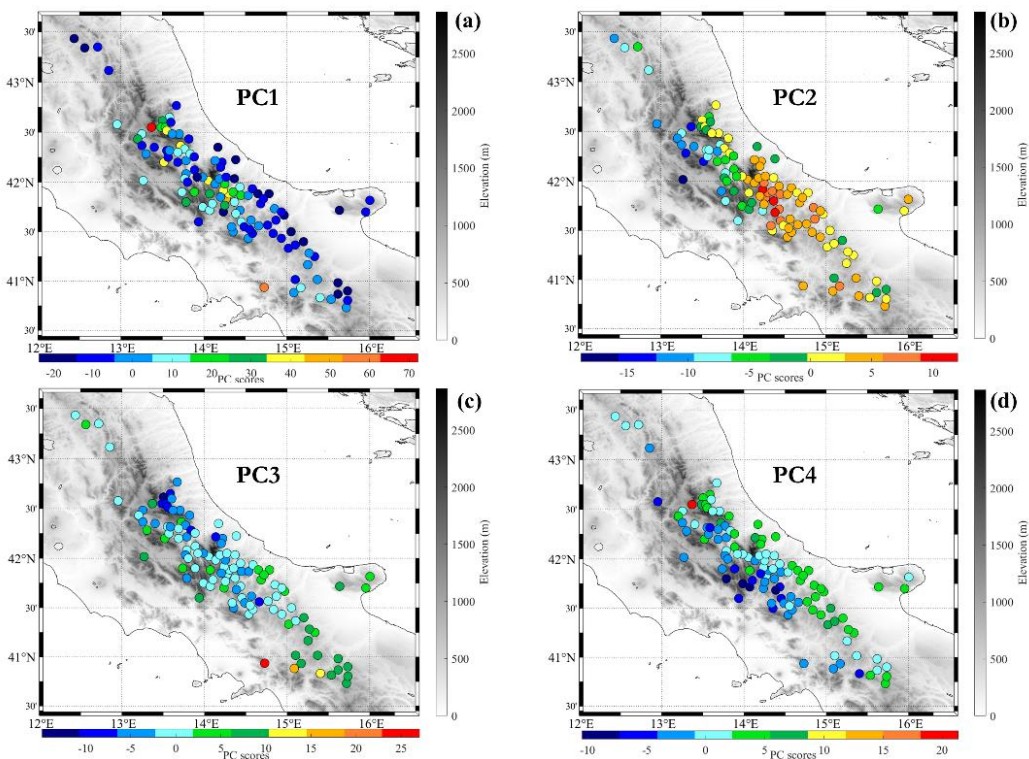

**Fig. B3. Spatial patterns of the first four modes resulting from the Principal Component Analysis applied to monthly NDS data.**

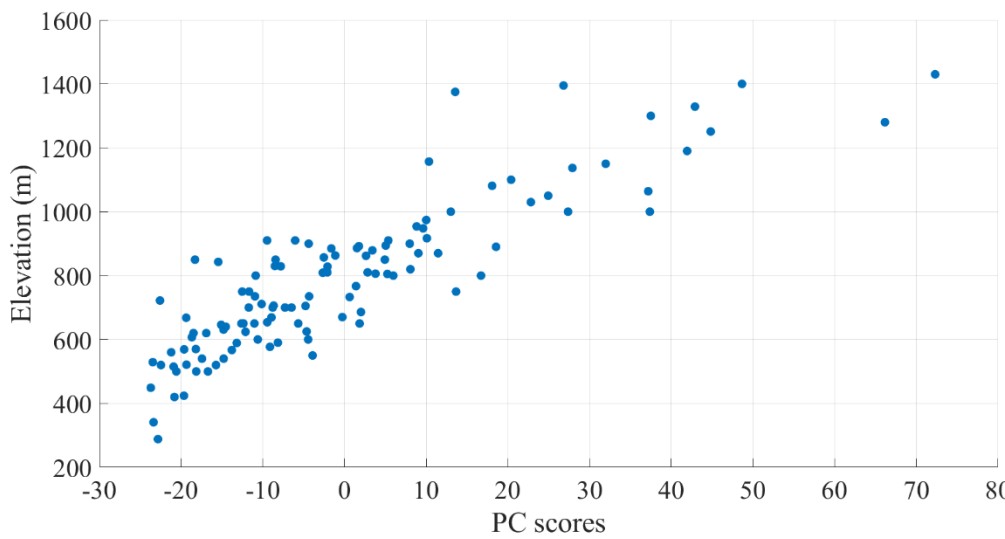


**Fig. B4. First principal component (PC1) scores resulting from PCA applied to monthly NDS data as function of the elevation (in m). Each point represents one station.**

The other five selected PCs are presented in Fig. B5. It is worth noting that such spatial patterns represent a very small fraction of variability (2% for PC5, PC6, PC7 and PC8, and 1% for PC9), so it is not straightforward identifying a "coherent" behaviour

in the spatial distribution of the scores. More specifically, in the PC5 spatial pattern (Fig. B5a), the most relevant positive NDS anomalies occurred in the Gran Sasso area (northern of Abruzzo) and in the Campania Apennine (Partenio mountains). PC6 pattern (Fig. B5b) is close to PC5: however, in this case positive scores, and so positive NDS anomalies, are confined to the Marsicani mountains area. The PC7 spatial pattern (Fig. B5c) reflect meteorological scenarios that determine positive NDS anomalies over the central and northern sectors of Abruzzo region, Molise and Campania Apennine. In PC8 spatial pattern

(Fig. B5d), positive scores are confined to specific sector of Abruzzo (Gran Sasso and Marsicani mountains) and to the southern sector of Molise. Finally, in PC9 the highest scores are located over the Gran Sasso area, Molise region and, locally, over the Campania Apennine (Fig. B5e).

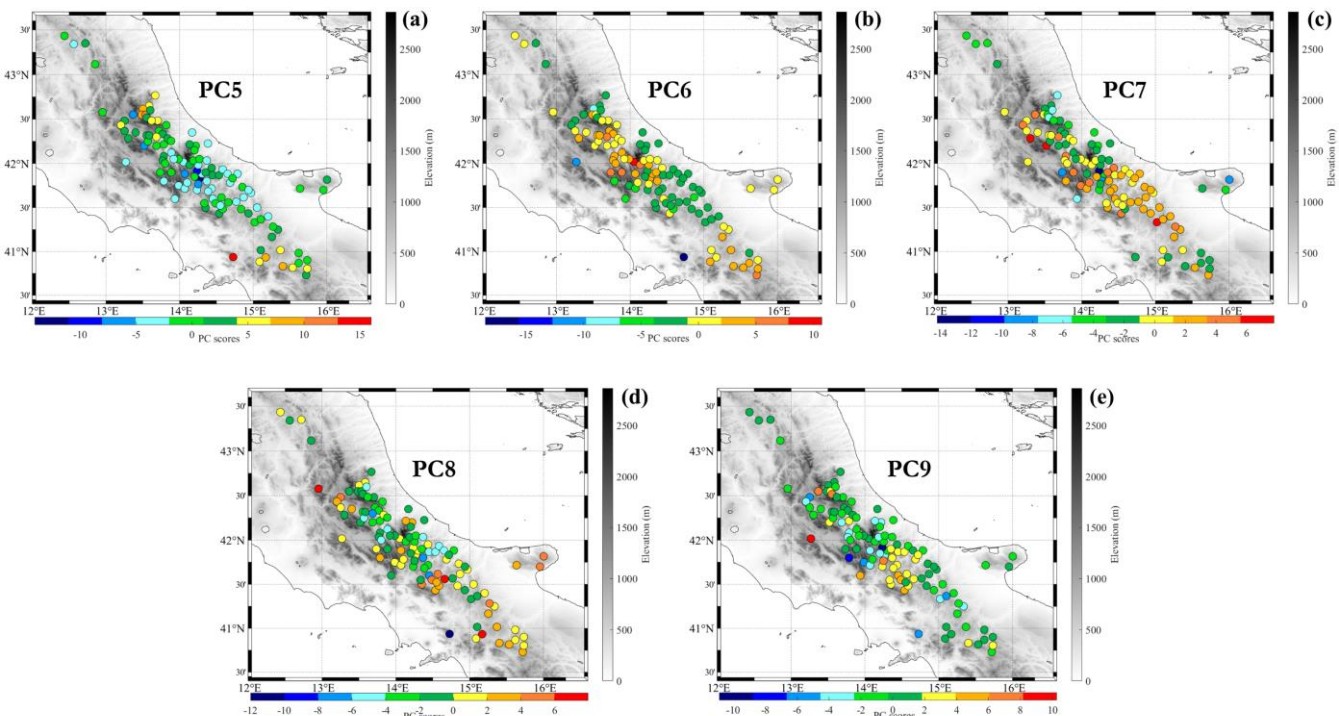

**Fig. B5. Spatial patterns of the fifth, sixth, seventh, eighth and ninth modes resulting from the Principal Component Analysis applied**
**to monthly NDS data.**

The results for height of new snow variable (HN) are presented in Fig. B6. Similarly to SCD, the first four PCs have been selected. The first PC, accounting for the 52% of the total variance, shows a spatial pattern strongly modulated by the altitude (Fig. B6a). As for SCD and NDS, a strong positive correlation between scores and elevation has been detected ($\rho = 0.83$). However, in this case the scores associated to stations above 800 m ASL exhibit a great variability (see Fig. B7), due to the
relevant incidence of orographic effects on snowfall amounts.

The analysis of PC2 spatial pattern (Fig. B6b) reveals a clear west-east gradient in the Central Apennine area. The large positive scores found over Majella area, Marsicani mountains, Matese and most of the Southern Apennine indicate that such areas

receive snowfall amounts substantially higher than average, whereas the negative scores over western side of Apennines are synonymous of HN quantity near or below average. This spatial pattern can be interpreted as a result of large-scale configurations that promote the incoming of cold continental air masses in the Central Mediterranean area. In this scenario, the Central and Southern Italy are often affected by a cyclonic area driving a north-eastern flow, which enhances orographic precipitation events over the eastern slopes of Apennines.

In the PC3 spatial pattern (Fig. B6c), the positive scores are concentrated over the Southern Apennine, in some areas of Molise and in the Reatini Mountains. In the Abruzzo region, the scores are generally negative, instead. Finally, the PC4 spatial pattern (Fig. B6d) is characterized by large positive scores over the western side of Marsicani area and the Reatini Mountains. In both PC3 and PC4, areas marked with positive scores receive snowfall amounts higher than average. Such spatial patterns can be related to specific large-scale weather patterns that modulate the spatial distribution of snow precipitation in the considered region.

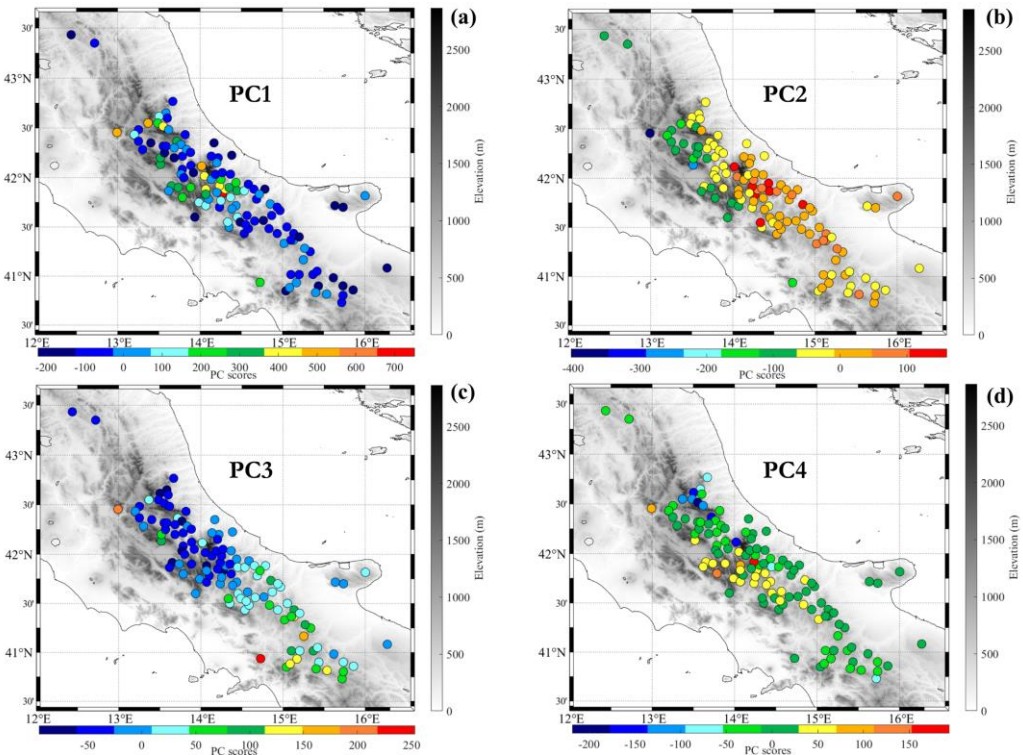

**Fig. B6. Spatial patterns of the first four modes resulting from the Principal Component Analysis applied to monthly HN data.**

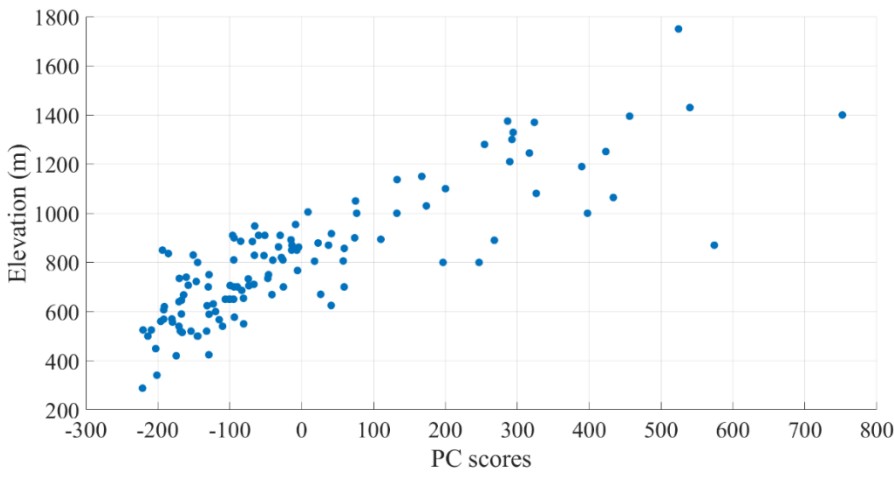


**Fig. B7. First principal component (PC1) scores resulting from PCA applied to monthly HN data as function of the elevation (in m). Each point represents one station.**


**Code/Data Availability**: The dataset that supports the findings of this study can be accessed through the following link: https://zenodo.org/records/12699507.

**Competing interests**. The authors declare that they have no conflict of interests.

**Author contribution**: Conceptualization, V.C.; Data curation, F.S., A.R. and V.C.; Methodology, V.C. and C.A.; Formal analysis, V.C. and F.S.; Investigation, V.C. and C.A.; Writing—original draft preparation, V.C.; Writing—review and editing, C.A., A.R. and G.B.; Supervision, G.B. All authors have read and agreed to the published version of the manuscript.

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

# LIST OF TABLES

**Table 1.** Main settings used to run Climatol (Guijarro, 2018) for quality control and homogenization of the investigated SCD, NDS and HN time series.

| Parameter | SCD | NDS | HN | Description |
|---|---|---|---|---|
| std | 1 | 1 | 2 | Type of data normalization: 1 = deviation from the mean; 2 = ratios to the mean |
| wz | 0.1 | 0.1 | 0.1 | Scale parameter of the vertical coordinate |
| dz.max | 15 | 15 | 20 | Threshold of outlier tolerance, in standard deviations |
| snht1 | 25 | 32 | 30 | Detection thresholds of changes in the mean of the series |
| snht2 | 40 | 40 | 40 | (determined by the explanatory analysis) |


**Table 2.** For each of the investigated snow indicators (SCD, NDS and HN), the number of stations belonging to a determined cluster and the stations' median and range elevation (minimum-maximum) are shown.

| Cluster | Number of stations | | | Elevation (m ASL) | | |
|---|---|---|---|---|---|---|
| | SCD | NDS | HN | SCD | NDS | HN |
| 1 | 58 | 31 | 52 | 648 (288-910) | 569 (288-850) | 603 (288-948) |
| 2 | 26 | 34 | 45 | 815 (550-954) | 700 (520-910) | 829 (625-1137) |
| 3 | 14 | 35 | 23 | 1000 (750-1157) | 879 (650-1375) | 1210 (800-1750) |
| 4 | 12 | 14 | / | 1290 (1000-1430) | 1220 (1000-1430) | / |



**Table 3.** For each cluster (CL) and for each season (full (F), early (E), winter (W) and late (L)), the average SCD trend value (expressed as number of days per 10 years), the percentage of positive and negative trend and the percentage of no trend are presented. For both positive and negative trend, the fraction of significant tendencies is also indicated. Note that no trend are defined as tendencies ranging between −0.4 and 0.4 days/10 years. The trend confidence level is coded as follows: **for 95%, * for 90%.

| CL | Average SCD trend (days/10 years) | | | | Positive trends (%) | | | | | | | | Negative trends (%) | | | | | | | | No trend (%) | | | |
|---|---|---|---|---|---|---|---|---|---|---|---|---|---|---|---|---|---|---|---|---|---|---|---|---|
| | | | | | Tot. | | | | Sig. | | | | Tot. | | | | Sig. | | | | | | | |
| | F | E | W | L | F | E | W | L | F | E | W | L | F | E | W | L | F | E | W | L | F | E | W | L |
| 1 | −1.1* | −0.4 | −1.1* | −0.3 | 3 | 2 | 0 | 0 | 0 | 0 | 0 | 0 | 72 | 43 | 81 | 28 | 21 | 14 | 24 | 7 | 24 | 55 | 19 | 72 |
| 2 | −1.3 | −0.6 | −1.7* | −0.3 | 8 | 8 | 0 | 8 | 0 | 0 | 0 | 0 | 73 | 73 | 86 | 62 | 27 | 12 | 35 | 4 | 19 | 19 | 12 | 31 |
| 3 | −3.0 | −1.1 | −2.8** | −1.4 | 0 | 7 | 0 | 0 | 0 | 0 | 0 | 0 | 86 | 86 | 93 | 86 | 36 | 36 | 50 | 7 | 14 | 7 | 7 | 14 |
| 4 | −3.4* | −1.1 | −3.2* | −1.8 | 8 | 17 | 0 | 8 | 0 | 0 | 0 | 0 | 92 | 67 | 92 | 83 | 42 | 17 | 50 | 42 | 0 | 17 | 8 | 8 |

**Table 4.** For each cluster (CL) and for each season (full (F), early (E), winter (W) and late (L)), the average NDS trend value (expressed as number of days per 10 years), the percentage of positive and negative trend and the percentage of no trend are presented. For both positive and negative trend, the fraction of significant tendencies is also indicated. Note that no trend are defined as tendencies ranging between −0.4 and 0.4 days/10 year. The trend confidence level is coded as follows: **for 95%, * for 90%.

| CL | Average NDS trend (days/10 years) | | | | Positive trends (%) | | | | | | | | Negative trends (%) | | | | | | | | No trend (%) | | | |
|---|---|---|---|---|---|---|---|---|---|---|---|---|---|---|---|---|---|---|---|---|---|---|---|---|
| | | | | | Tot. | | | | Sig. | | | | Tot. | | | | Sig. | | | | | | | |
| | F | E | W | L | F | E | W | L | F | E | W | L | F | E | W | L | F | E | W | L | F | E | W | L |
| 1 | −0.8** | −0.3 | −0.6** | −0.3 | 0 | 0 | 0 | 0 | 0 | 0 | 0 | 0 | 71 | 32 | 71 | 13 | 48 | 29 | 52 | 0 | 29 | 68 | 29 | 87 |
| 2 | −0.6 | −0.3 | −0.7** | −0.1 | 0 | 0 | 0 | 0 | 0 | 0 | 0 | 0 | 65 | 32 | 79 | 14 | 32 | 24 | 29 | 3 | 35 | 68 | 21 | 85 |
| 3 | −1.2** | −0.6* | −1.2** | −0.5 | 0 | 0 | 0 | 0 | 0 | 0 | 0 | 0 | 89 | 83 | 97 | 60 | 60 | 46 | 69 | 29 | 11 | 17 | 3 | 40 |
| 4 | −1.7** | −0.9* | −1.6** | −0.6 | 0 | 0 | 0 | 0 | 0 | 0 | 0 | 0 | 93 | 86 | 100 | 79 | 71 | 71 | 79 | 14 | 7 | 14 | 0 | 21 |

# LIST OF FIGURES

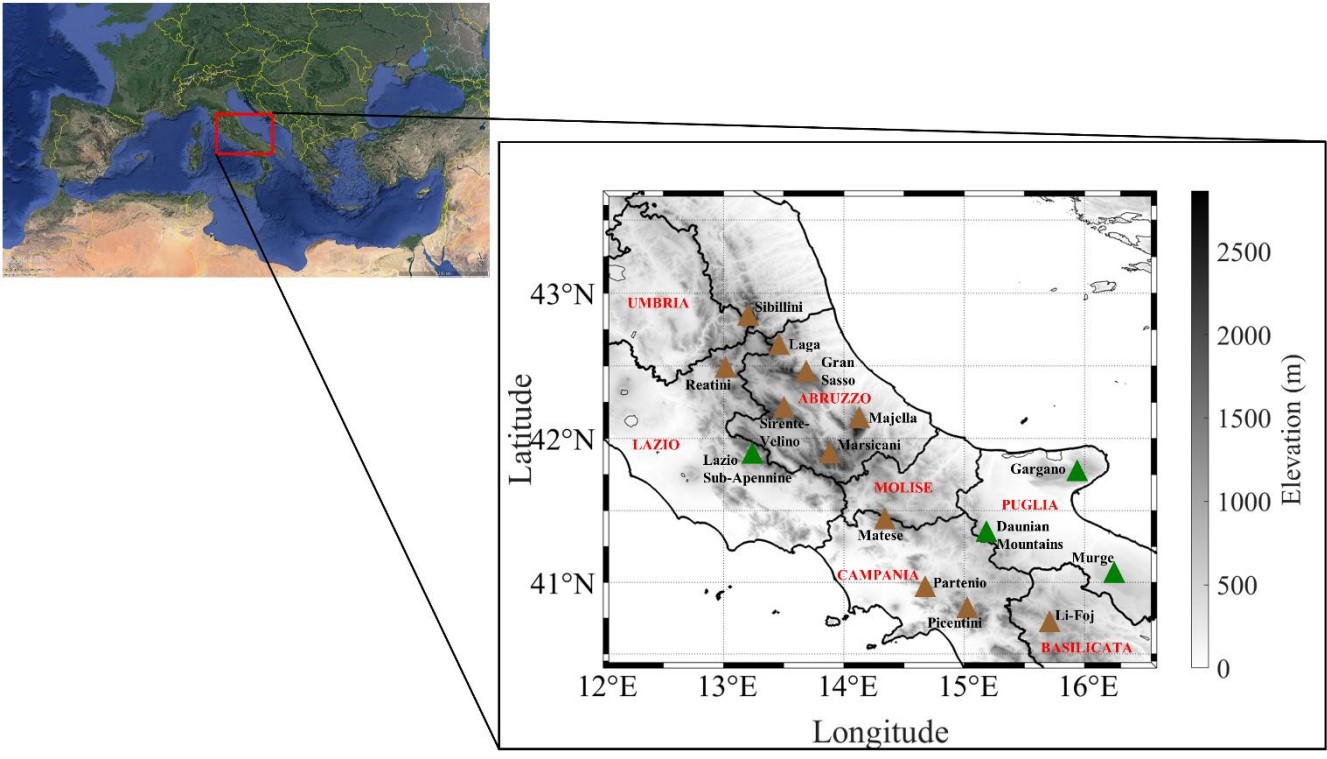

**Figure 1: In the left panel, a map of Mediterranean area, including the study region (highlighted as red solid-outlined box) is presented. The right panel shows a digital elevation model of the investigated area, with several mountain ranges mentioned in the main text. More specifically, the main Apenninic reliefs are marked as brown filled-in triangles, whereas the filled-in green triangles indicate several Apennine offshoots. The black line shows the boundaries of the Italian administrative regions included in the study area (the official name of the regions is indicated in red). Images credit: © Google Earth, Data Sio, NOAA, U.S. Navy, NGA, GEBCO.**

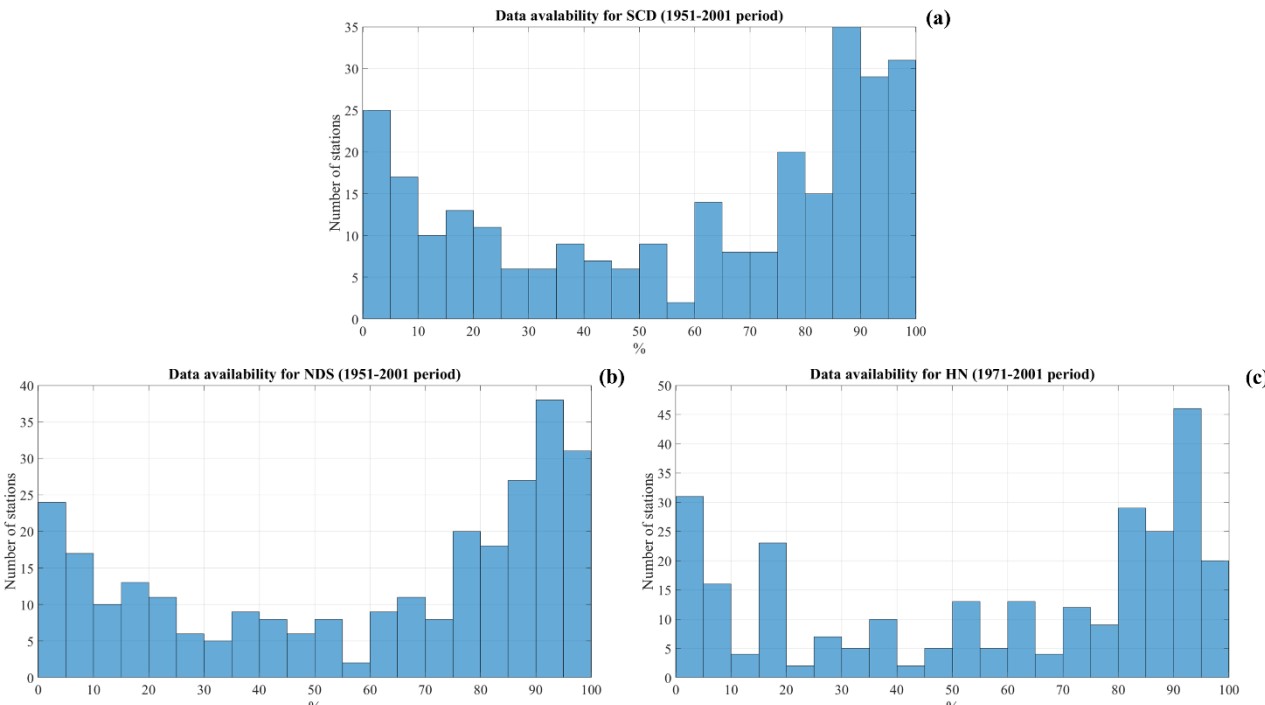

Figure 2: Histogram of the data availability (in %) for the considered snowfall parameters: SCD (a), NDS (b) and HN (c). Note that in this figure all rescued stations have been considered.

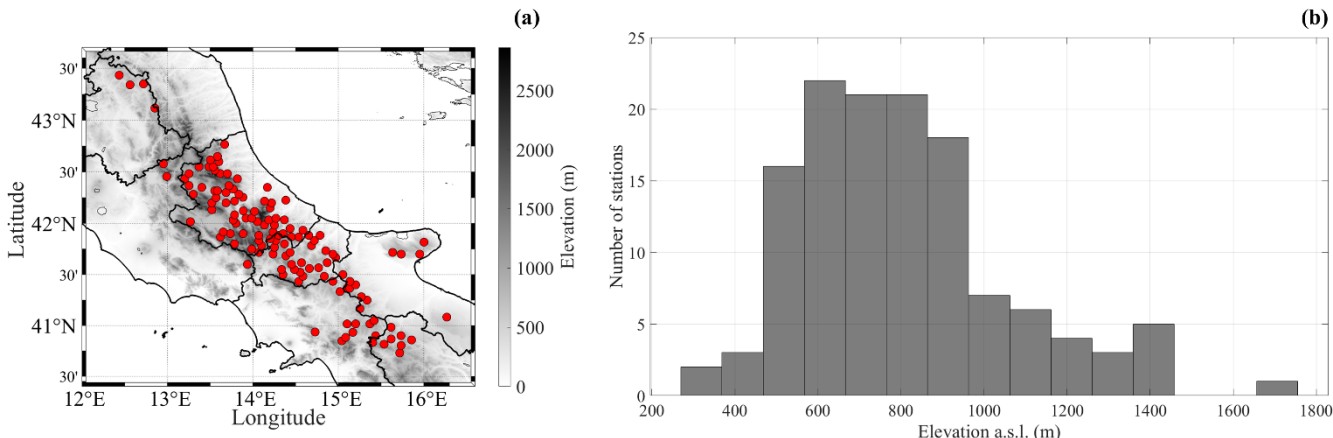

Figure 3: Panel (a) shows the location of the stations (129 in total) considered for the analyses carried out in this work. The black line shows the boundaries of the Italian administrative regions included in the study area. Panel (b) sketches the number of available stations per elevation.

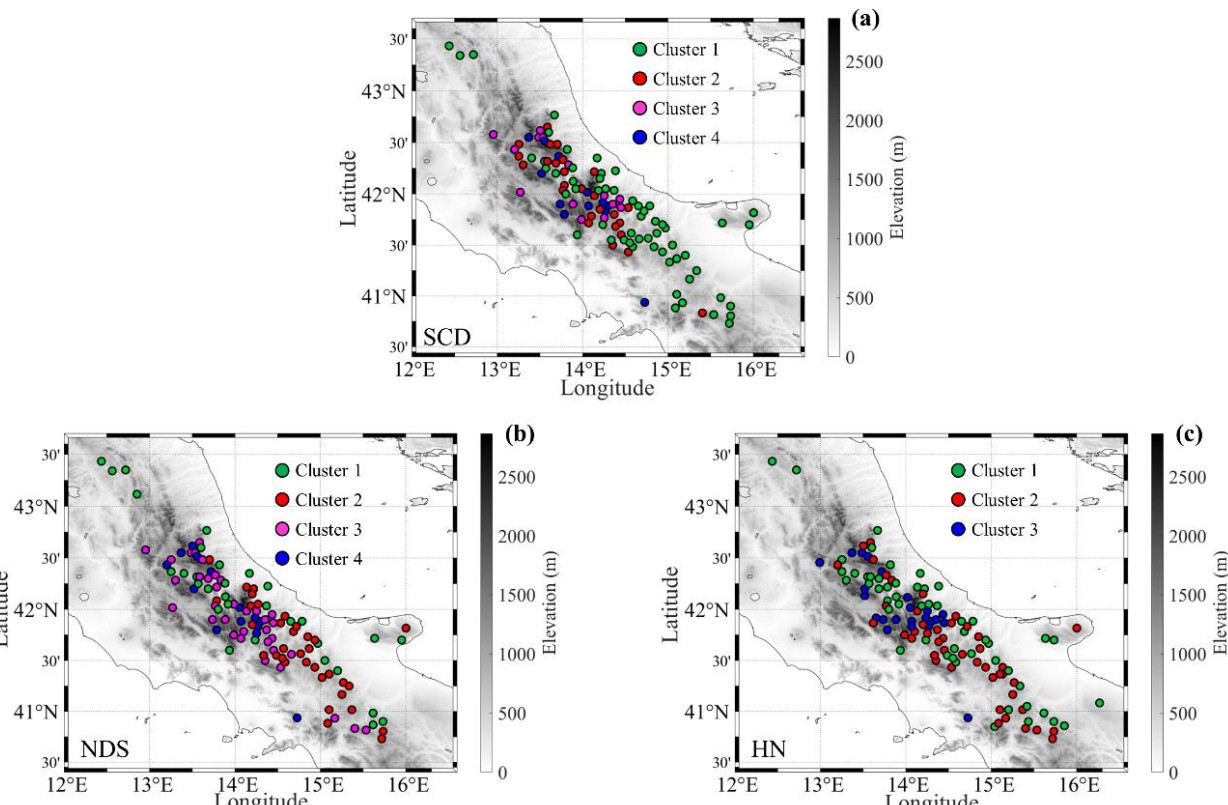

**Figure 4: Clustering of stations based on monthly data of SCD (a), NDS (b) and HN (c). The stations are color-coded according to the cluster memberships.**

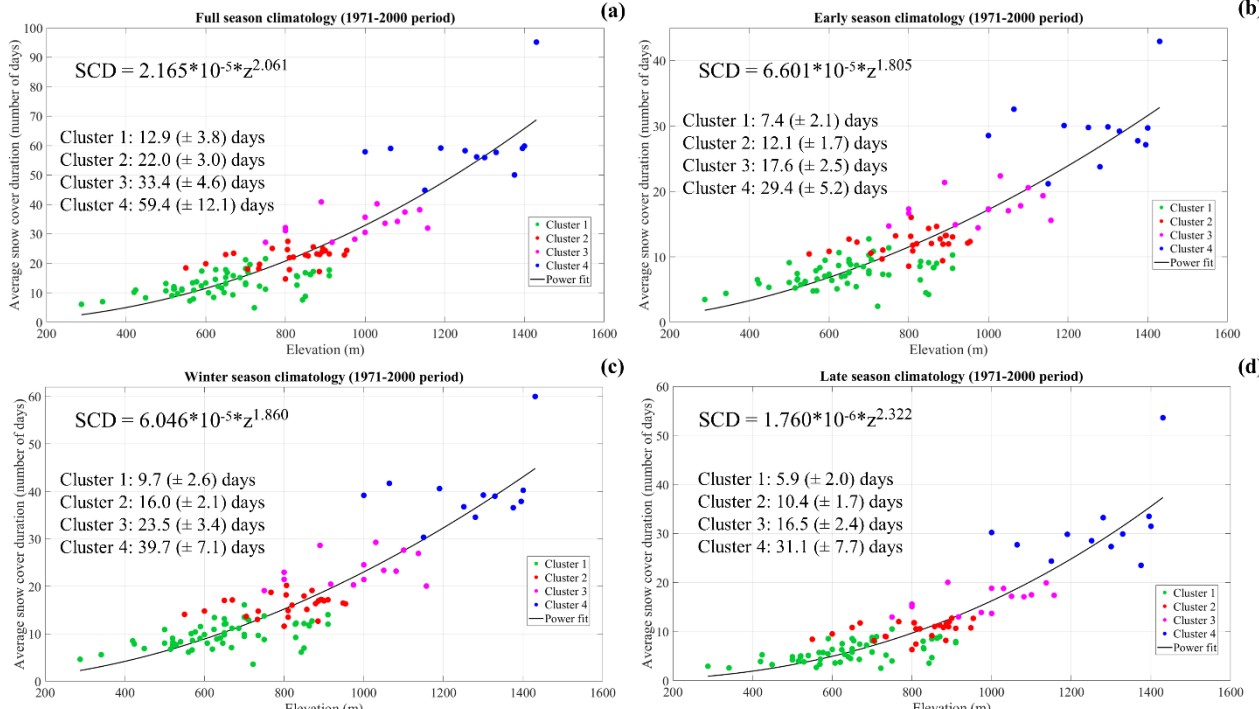

**Figure 5: Climatology of snow cover duration (SCD) for (a) full, (b) early, (c) winter and (d) late season. Average values are for the period 1971-2000. Each point represents a station that is color-coded according to the membership cluster. The black solid line represents the power fit. The text boxes show the power fit equation and the average and standard deviation values for each cluster.**


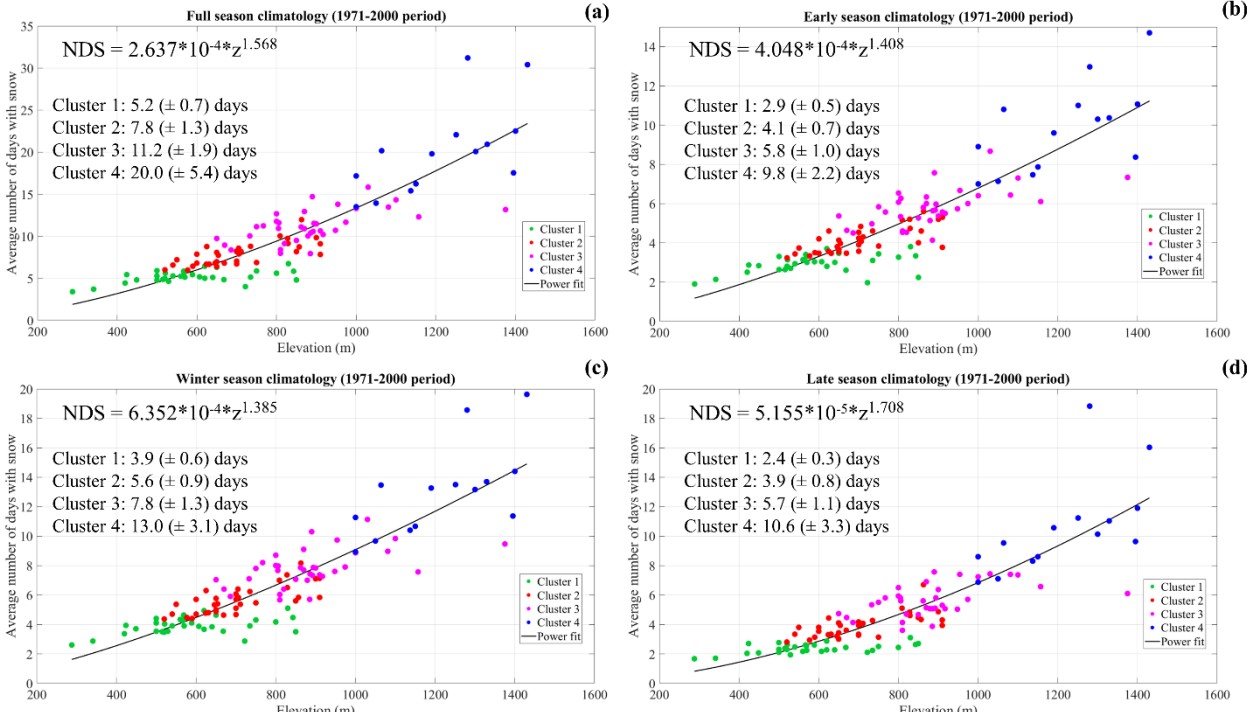

**Figure 6: Climatology of number of days with snow (NDS) for (a) full, (b) early, (c) winter and (d) late season. Average values are for the period 1971-2000. Each point represents a station that is color-coded according to the membership cluster. The black solid 1095 line represents the power fit. The text boxes show the power fit equation and the average and standard deviation values for 1285 each cluster.**

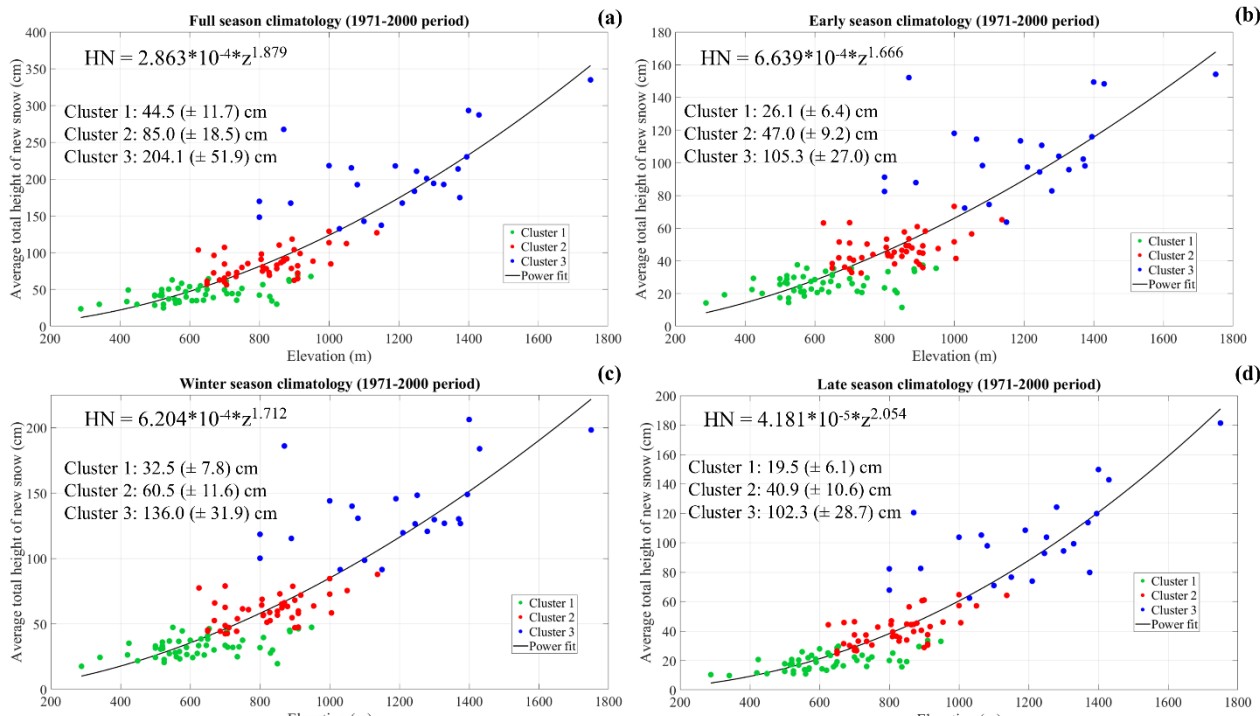

**Figure 7: Climatology of height of new snow (HN) for (a) full, (b) early, (c) winter and (d) late season. Average values are for the period 1971-2000. Each point represents a station that is color-coded according to the membership cluster. The black solid line represents the power fit. The text boxes show the power fit equation and the average and standard deviation values for each cluster.**


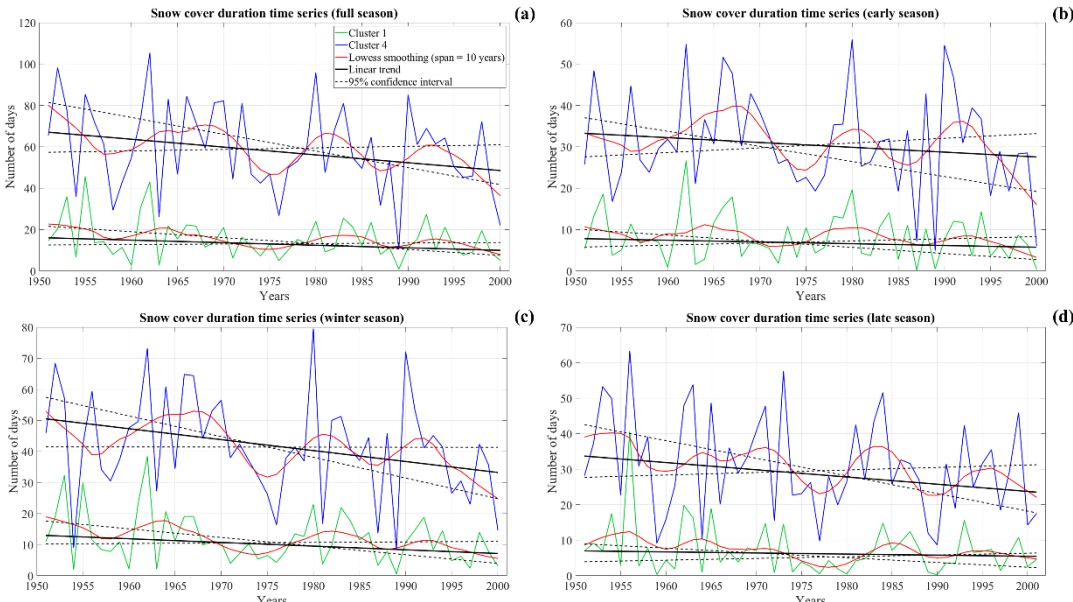

**Figure 8: Cluster 1 (green line) and cluster 4 (blue line) snow cover duration (SCD) time series for full (a), early (b), winter (c) and late (d) season. The black solid line shows the linear trend, the dashed black line the 95% confidence interval for linear trend, whereas the red line marks the lowess smoothing (span = 10 years). The period from 1951 to 2001 has been considered.**


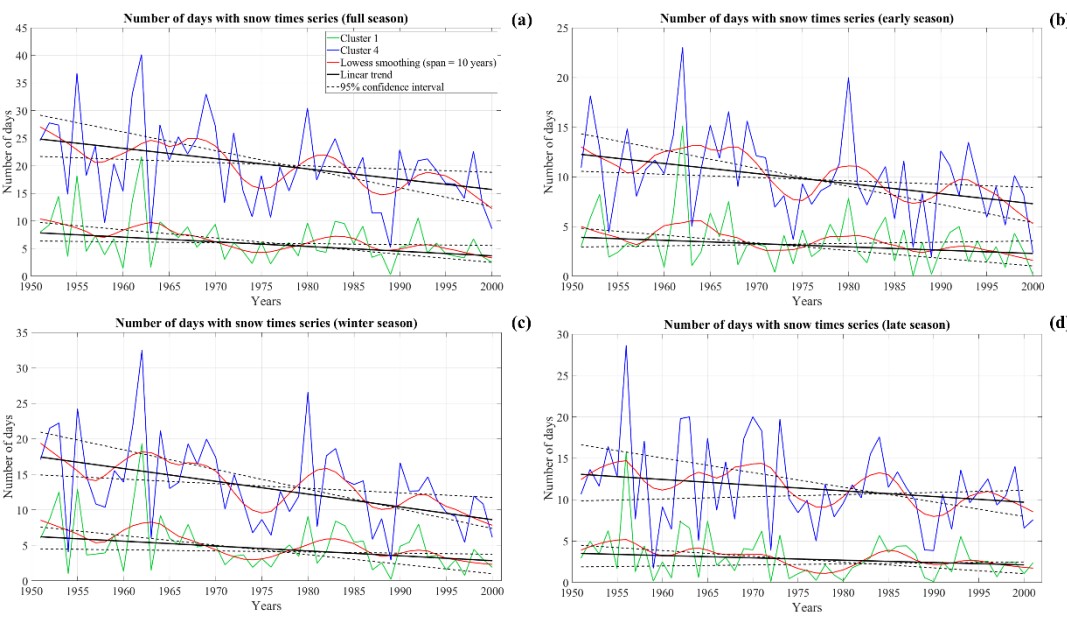

**Figure 9: Cluster 1 (green line) and cluster 4 (blue line) number of days with snow (NDS) time series for full (a), early (b), winter (c) and late (d) season. The black solid line shows the linear trend, the dashed black line the 95% confidence interval for linear trend, whereas the red line marks the lowess smoothing (span = 10 years). The period from 1951 to 2001 has been considered.**


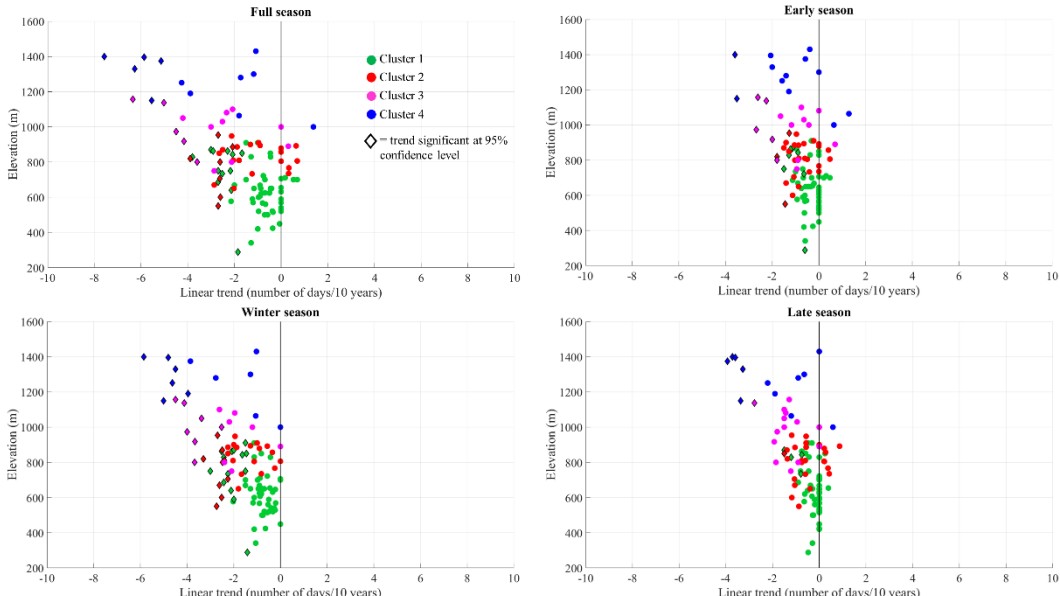

**Figure 10: Seasonal snow cover duration (SCD) trends (expressed as number of days/10 years) as function of elevation. Each point represents one station and is color-coded according to the membership cluster. Trends significant at 95% confidence level are displayed as diamond marker.**


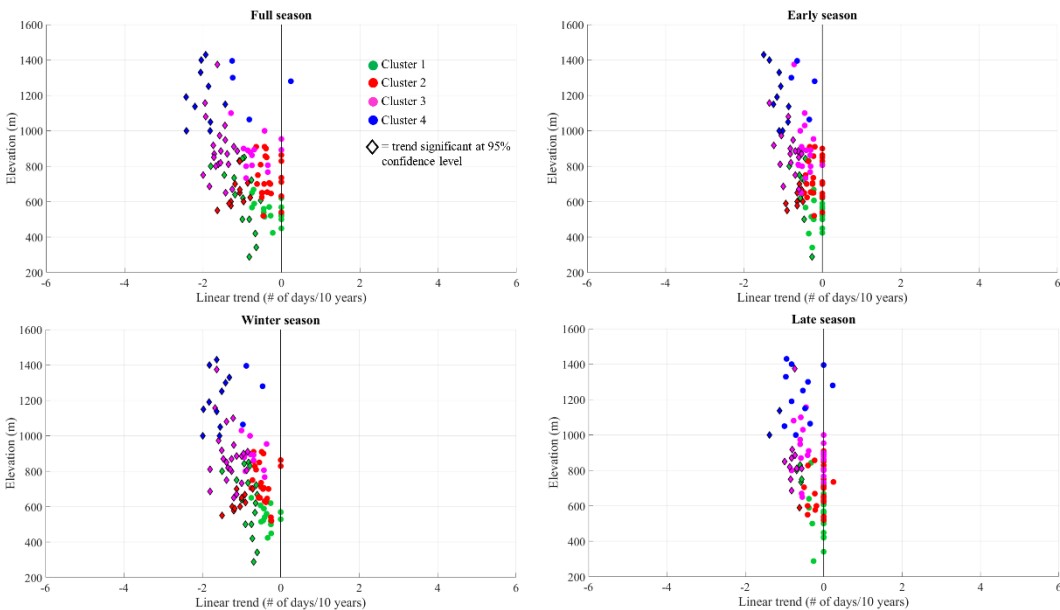

**Figure 11: Seasonal number of days with snow (NDS) trends (expressed as number of days/10 years) as function of elevation. Each point represents one station and is color-coded according to the membership cluster. Trends significant at 95% confidence level are displayed as diamond marker.**

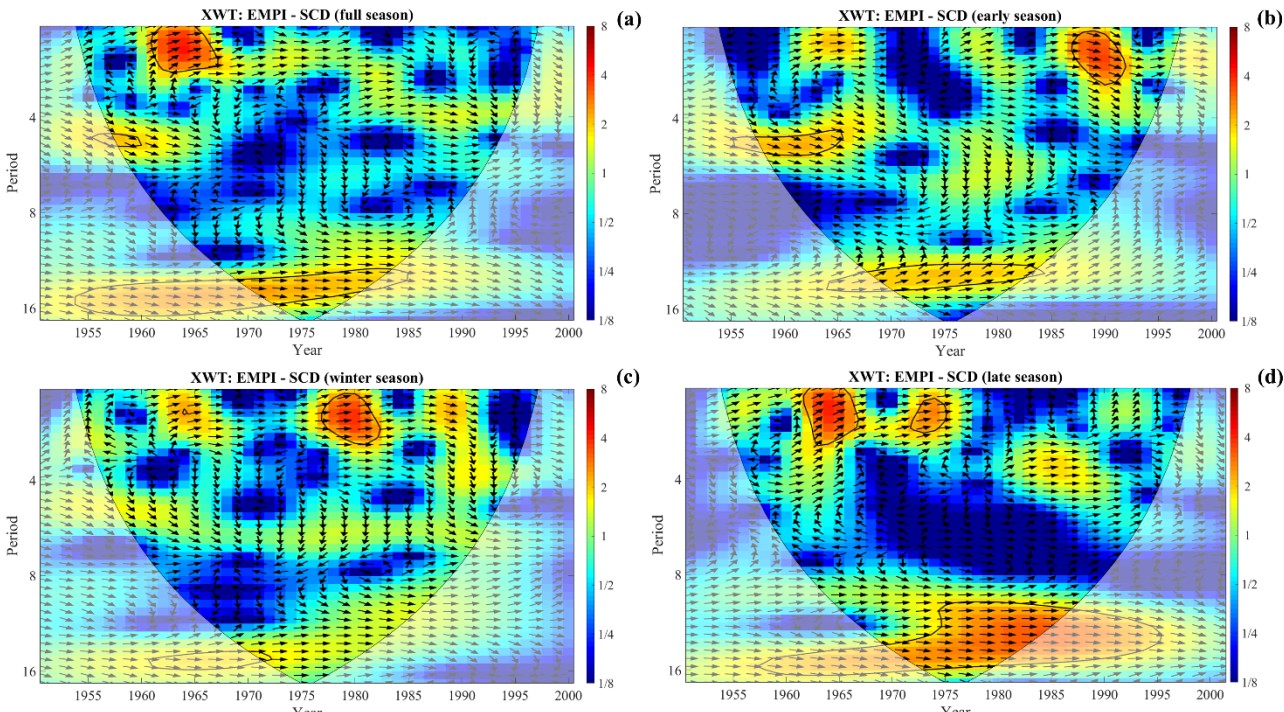


**Figure 12: Cross Wavelet Transform (XWT) between Eastern Mediterranean Pattern Index (EMPI) and the cluster 4 snow cover duration (SCD) time series for (a) full season, (b) early season, (c) winter season, and (d) late season. The black arrows indicate the phase relationship between the respective time series. The thick contour designates the 5% significance level against red noise; the cone of influence, where edge effects might distort the picture, is shown as a lighter shade. All XWT spectra refer to the period 1951-**
**2001. Note that black arrows pointing to the right indicate that the signals are in phase.**

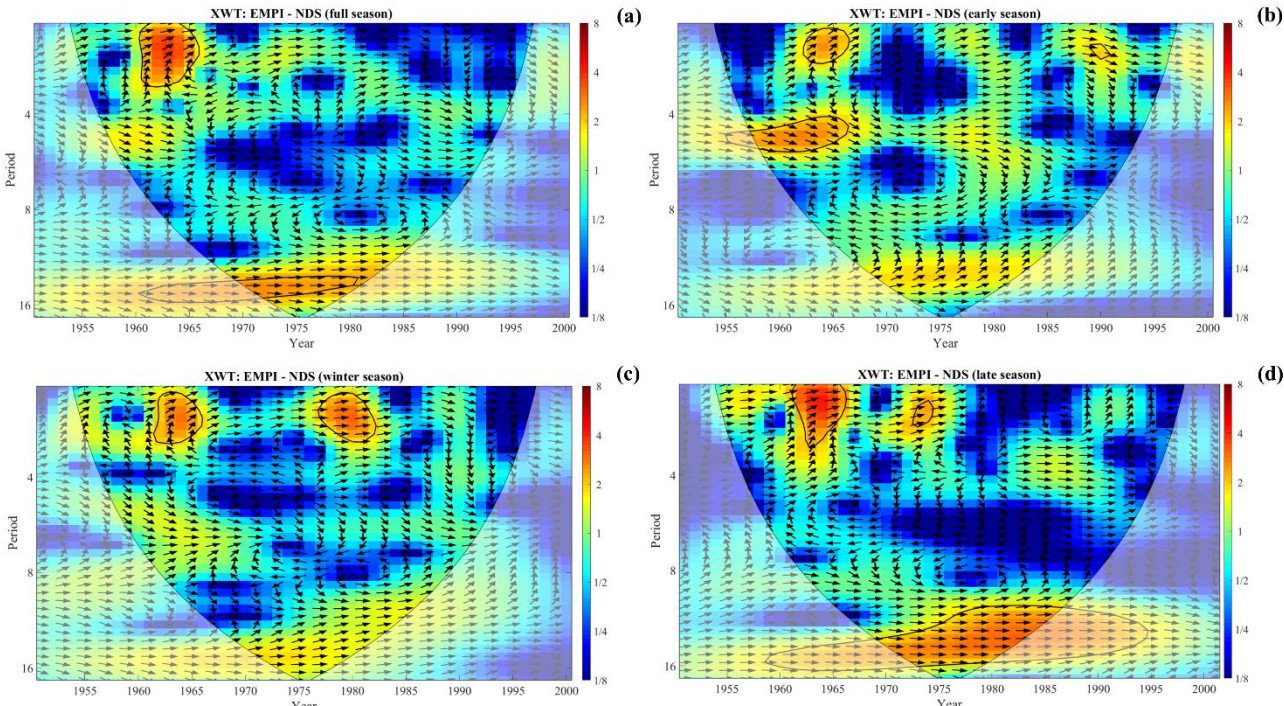

**Figure 13: Cross Wavelet Transform (XWT) between Eastern Mediterranean Pattern Index (EMPI) and the cluster 4 number of days with snow (NDS) time series for (a) full season, (b) early season, (c) winter season, and (d) late season. The black arrows indicate the phase relationship between the respective time series. The thick contour designates the 5% significance level against red noise; the cone of influence, where edge effects might distort the picture, is shown as a lighter shade. All XWT spectra refer to the period 1951-2001. Note that black arrows pointing to the right indicate that the signals are in phase.**


