# Peer review of "Historical snow measurements in the Central and Southern Apennine Mountains: climatology, variability and trend"

_EGUsphere, 2024_

## Author Comment (AC2)

Revision of

**"Historical snowfall measurements in the Central and Southern Apennine Mountains: climatology, variability and trend"**

RC (X) = Referee comment (number X)
AR (X) = Authors' reply (number X)

**REVIEWER #1**

**AC:** Dear Referee, we are grateful for the time dedicated to the revision of our manuscript and for the suggestions, which help us to improve our paper. Here we provide a point-by-point response to his/her comments. All required changes will be included in the new manuscript version.

**RC (1):** This paper presents a valuable collation of historical snow records for an understudied region. As such, the authors should fully document and deposit the data in a public repository, in compliance with the Copernicus Publications data policy: https://www.the-cryosphere.net/policies/data_policy.html
**AC (1):** Following this valuable suggestion, we have deposited the dataset that supports this study in the Zenodo open data repository (CERN). The dataset can be accessed through the following link: https://zenodo.org/records/12699507

**RC (2):** The cluster analysis is detailed, but I am not sure that the discussion reveals much more than could have been illustrated by plotting the snow metrics against elevation and examining the outliers. Although the wavelet analysis is rather preliminary and descriptive, it presents results and so should be moved to the Results section.
**AC (2):** Following your recommendation, in new manuscript we'll move the wavelet analysis in the Results section.

**RC (3):** 15
The difference between "snow cover duration" and "number of days with snow" is not clear, and is not made clear until line 183. State "number of days with snowfall" throughout.
**AC (3):** Ok, thank you for the suggestion.

**RC (4):** 31
Snowfall is certainly an essential climate variable, but it is not a GCOS Essential Climate Variable distinct from precipitation, so do not use that specific term.
**AC (4):** Ok, in the new manuscript version we'll remove this term.

**RC (5):** 257-300
The description of Climatol tests is barely comprehensible without reading the references.
**AC (5):** Following this suggestion, we have revised the description of Climatol test. Here we provide the new description. Note that the changes with respect to the original manuscript version are highlighted in yellow.

[revised manuscript text omitted]

**RC (6):** 360
Relationships of PCs to geographical features are stated but not made clear to the reader.
**AC (6):** Thank you for this comment. Here we provide a detailed description of the Principal Component Analysis (PCA) results for each of the three investigated variables: snow cover duration (SCD), number of days with snowfall (NDS) and height of new snow (HN). Such analysis will be included in the new version of the manuscript as Appendix B. It is important highlighting that the aim of our PCA analysis is to identify the dominant recurring spatial patterns over time of the investigated parameters. A natural evolution of this type of analysis is the research of the atmospheric circulation characteristics that are associated with the identified spatial structures of SCD, NDS and HN variables. This aspect is very interesting and fits well with our research interests. However, it falls out the scope of this paper, so it will be addressed in future work.

For SCD, we have selected the first four Principal Components (PCs), which account for the 75% of the total variance. Fig. 1 of this document shows the spatial pattern of the PC scores. Please consider Fig. 1 of the original manuscript for locations mentioned therein. The first PC (Fig. 1a), 
[revised manuscript text omitted]

**RC (7):** 360
There is some appeal to having elevation on the y-axis, but it would conventional for it to be on the x-axis as the independent variable. This would also better show the overlap in elevation between clusters and the increasing gradient. Rather than the generic x = ay^b, it would be better to show the power fit equations as SCD = az^b.
**AC (7):** Following the valuable referee's suggestion, we have revised the Fig. 5, Fig. 6 and Fig. 7 of our manuscript. Here we present the new version of such figures. Note that the independent variable $z$ in the power fit equations stands for elevation (in m).

[Figure]

**Figure 5**: Climatology of snow cover duration (SCD) for (a) full, (b) early, (c) winter and (d) late season. Average values are for the period 1971-2000. Each point represents a station that is color-coded according to the membership cluster. The black solid line represents the power fit. The text boxes show the power fit equation and the average and standard deviation values for each cluster.

[Figure]

**Figure 6:** Climatology of number of days with snow (NDS) for (a) full, (b) early, (c) winter and (d) late season. Average values are for the period 1971-2000. Each point represents a station that is color-coded according to the membership cluster. The black solid 1095 line represents the power fit. The text boxes show the power fit equation and the average and standard deviation values for each cluster.

[Figure]

**Figure 7:** Climatology of height of new snow (HN) for (a) full, (b) early, (c) winter and (d) late season. Average values are for the period 1971-2000. Each point represents a station that is color-coded according to the membership cluster. The black solid line represents the power fit. The text boxes show the power fit equation and the average and standard deviation values for each cluster.

**RC (8):** 464
"subset" would be a more widely comprehensible term than "aliquot".
**AC (8):** Ok, thank you.

**RC (9):** Figure 11
Does this contradict recovery of NDS in the Southern Apennines cited in the introduction?
**AC (9):** According to Capozzi et al. (2022) and Annella et al. (2023), a recovery in NDS has been observed, in the Southern Apennine area, in the last 20 years (i.e. in the 2000-2020 period). Note that this evidence emerged from the analysis of Montevergine time series, which is one of the few historical series that extends up to recent years. In our study, for the reason explained in Section 2.2, we focused on the 1951-2001 period. Therefore, the results sketched in Figure 11 are not in contradiction with the NDS recovery mentioned in the Introduction (i.e. the recovery occurred after 2000).

**RC (10):** 593
XX century?
**AC (10):** Yes, XX century, sorry for the mistake.

**RC (11):** Figures 12 and 13
The captions should state that arrows pointing to the right indicate that signals are in phase.
**AC (11):** Ok, we'll follow this suggestion. Thank you.

**RC (12):** 675
When claiming 90% confidence, it would make more sense to quote the 90% confidence interval.
**AC (12):** Ok, thank you for this suggestion.

---

## Author Comment (AC3)

Revision of

**"Historical snowfall measurements in the Central and Southern Apennine Mountains: climatology, variability and trend"**

RC (X) = Referee comment (number X)
AC (X) = Authors' reply (number X)

**REVIEWER #2**

**RC:** Dear Authors, dear Editor,
Thank you for proposing this study. The topic is interesting and timely, the methods used are sound, and the focus on the Appenine area is a valuable complement to pre-existing work and snow data analyses.
Therefore I find the presented work very valuable and worth publishing - but naturally, I have some suggestions to try to improve it.
**AC:** Dear Referee, we are very grateful for the positive evaluation of our manuscript and for the comments and the suggestions, which help us to improve our paper. Here we provide a point-by-point response to his/her comments. All required changes will be included in the new manuscript version.

**RC (1):** I join referee one in his/her concern that the data should be made openly accessible (for instance via a doi associated to the present paper) to comply with Copernicus guidelines.
**AC (1):** Following this valuable suggestion, we have deposited the dataset that supports this study in the Zenodo open data repository (CERN). The dataset can be accessed through the following link: https://zenodo.org/records/12699507

**RC (2):** As also assessed by referee 1, "Number of days with Snow" / NDS is too vague (notably L55) and the description comes too late in the paper. As I understand from L 183 it could be formulated as Number of days with fresh snow accumulation on the ground.
**AC (2):** In the new manuscript version, "Number of days with snow" will be replaced by "Number of days with snowfall". In addition, following your valuable comment, we will introduce more clearly this parameter at Line 55.

**RC (3):** The Standard Normal Homogeneity Test procedure is barely understandable the way it is currently presented without reading further reference. I suggest to explain the general philosophy underlying the test.
**AC (3):** Thank you for the suggestion. We have introduced the following brief description. Note that the changes with respect to the original manuscript version are highlighted in yellow.

"Climatol has been employed in this study also to check for homogeneity of the investigated time series. The use of this toolbox for the homogenisation of snowfall data has been explored, with encouraging results, in some recent works (Buchmann et al., 2022; Buchmann et al., 2023). As described in detail by Guijarro (2018) and by Kuya et al. (2022), the Climatol homogenization method is based on the Standard Normal Homogeneity Test (SNHT; Alexandersson, 1986) for the identification of the breaks and on a linear regression approach for the adjustments (Easterling and

Peterson, 1995). The SNHT falls within homogenization procedures that are able to identify an inhomogeneity without knowing a priori the time of the break point in the time series and that can also estimate the magnitude of the detected break. The basic idea underlying this method consists in using neighbouring stations as a reference to identify inhomogeneities in the station being tested (the candidate station). Such assumption requires the existence of a sufficient correlation level between test and reference stations. More specifically, SNHT uses normalised series of the ratios/differences (hereafter, $Q$) between e.g. precipitation/temperature at candidate station and neighbouring reference stations. The test is based on the null hypothesis that the $Q$ series has a constant mean level, i.e. that the candidate series is homogeneous, and the alternative hypothesis that the mean level of the $Q$ series changes abruptly from one level to another at some time. For each point of the time series, a test value, based on a comparison between the means of the two subsamples before and after the potential breakpoint, is computed as described in detail in Alexandersson and Moberg (1997). The null hypothesis is rejected if the maximum test value of all dividing points in the $Q$ series is greater than a predefined critical level. In Climatol, the SNHT is applied to the anomalies time series previously introduced in the description of the tolerance test."

**References**

Alexandersson, H., & Moberg, A. (1997). Homogenization of Swedish temperature data. Part I: Homogeneity test for linear trends. International Journal of Climatology: A Journal of the Royal Meteorological Society, 17(1), 25-34.

**RC (4):** More generally, the use of the terms "snow" is sometimes misleading throughout the paper, as illustrated in the expression "number of days with snow" . We don't know wether this is atmospheric snow (snowfall) or ground covering snow. Please be more specific.

**AC (4):** Ok, thank you for the suggestion. In the new manuscript version, we will specify it.

**RC (5):** Both the title and the abstract draw the focus on snow *precipitation* or *snowfall*. However, based on the variables analyzed (snow cover duration, number of days with snow, total height of new snow) the focus is at least equally on snow on the ground as on snowfall. This should be revised in order to convey a more precise message.

**AC (5):** Thank you for this valuable suggestion. To avoid ambiguity, we will revise the title of our manuscript, replacing "snowfall" with "snow". The term "snow" may be considered more general and inclusive of different types of data (e.g. snow cover, snow precipitation amount and snow frequency of occurrence) than "snowfall". As an example, Scherrer et al. (2013) have used a similar title ("Snow variability in the Swiss Alps, 1864-2009"), in a work that considered different snow indicators, such as new snow sums, maximum new snow and days with snowfall.

Therefore, in the new manuscript version we'll use the term "snow" to generally mention all the snow variables employed in our study and we will replace it with "snowfall" when referring a specific parameter, such as the "Number of days with snowfall".

**References**

Scherrer, S. C., Wüthrich, C., Croci-Maspoli, M., Weingartner, R., and Appenzeller, C.: Snow variability in the Swiss Alps 1864–2009, Int. J. Climatol., 33, 3162–3173, https://doi.org/10.1002/joc.3653, 2013.

Minor comments :

**RC (6):** L 43-44 : please aknowledge recent work that expands in time the MODIS time-series through the use of older satellite archives or reanalyses, and machine learning

Dumont, Z. B., Gascoin, S., & Inglada, J. (2024). Snow and cloud classification in historical SPOT images: An image emulation approach for training a deep learning model without reference data. IEEE Journal of Selected Topics in Applied Earth Observations and Remote Sensing.

Gascoin, S., Monteiro, D., & Morin, S. (2022). Reanalysis-based contextualization of real-time snow cover monitoring from space. Environmental Research Letters, 17(11), 114044.

**AC (6):** Ok, we will follow this suggestion. Thank you.

**RC (7):** L 172 : snow-to-liquid equivalent : the proper name of this quantity is snow water equivalent (SWE). See the International Classification of snow (Fierz et al., 2009) here: https://unesdoc.unesco.org/ark:/48223/pf0000186462

**AC (7):** Ok, thank you for the suggestion. In the new manuscript version, we will modify it.

**RC (8):** In one of the authors' previous work mentioned in the Introduction, a recent recovery of snow cover duration, HN and NDS is mentioned in the Southern Appenines (L 67 68). Unfortunately the present study ends by 2000 while the recovery at Montevergine Observatory appears after 2000. Would it be possible to bridge the gap between both studies and mention this in the discussion, enlightening the dependence of trends to timeframe/period length, and also the connection with AO/NA that was seen in this previous study but found not relevant in the present one?

**AC (8):** Ok, thank you for this suggestion. In the Discussion section, we will mention that previous studies on the Apennine region highlighted a recovery in snow cover duration, snowfall amounts and number of days with snowfall after 2000 and that this rebound in snow indicators is closely linked to the trend of Arctic Oscillation (AO). About the connections between AO and time series employed in this study, it is important pointing out that this aspect has been analysed only means of Cross Wavelet Transform. We feel that additional analyses are necessary to better assess the relationship between this relevant atmospheric mode and snow variability in the study area. It may speculate that non-negligible differences might exist between western and eastern sectors of Apennines (the first one might be more "sensitive" to the AO variability). In the Conclusions section, we will add a sentence about this future investigation.

Edits :

**RC (9):** L61 : clear -> clearly
**AC (9):** Ok, thank you.

**RC (10):** L 106 : ad -> an
**AC (10):** Ok, thank you.

**RC (11):** L 115 : southwest -> south east
**AC (11):** Ok, thank you.

**RC (12):** L 223 : quote -> quite
**AC (12):** Ok, thank you.

**RC (13):** L 263 : use -> uses
**AC (13):** Ok, thank you.

**RC (14):** L 355 : scree -> screen
**AC (14):** Ok, thank you.

**RC (15):** L 511 : second "in" -> and
**AC (15):** Ok, thank you.

**RC (16):** L 539 one word is missing (likely "to" before "contextualize").
**AC (16):** We are sorry for the error. We will add the missing word.

**RC (17):** L 559 : stations -> station
**AC (17):** Ok. Thank you.

**RC (18):** L 689 : means -> by means
**AC (18):** Ok, thank you.

---

## Author Response (AR1)

Revision of

**"Historical snowfall measurements in the Central and Southern Apennine Mountains: climatology, variability and trend"**

RC (X) = Referee comment (number X)
AR (X) = Authors' reply (number X)

**REVIEWER #1**

**AC:** Dear Referee, we are grateful for the time dedicated to the revision of our manuscript and for the suggestions, which help us to improve our paper. Here we provide a point-by-point response to his/her comments. Please note that all required changes have been marked in yellow in the new manuscript version.

**RC (1):** This paper presents a valuable collation of historical snow records for an understudied region. As such, the authors should fully document and deposit the data in a public repository, in compliance with the Copernicus Publications data policy: https://www.the-cryosphere.net/policies/data_policy.html
**AC (1):** Following this valuable suggestion, we have deposited the dataset that supports this study in the Zenodo open data repository (CERN). The dataset can be accessed through the following link: https://zenodo.org/records/12699507.
We have accordingly modified the Code/Data Availability Section of the manuscript (See Pag. 32, Lines 887-888).

**RC (2):** The cluster analysis is detailed, but I am not sure that the discussion reveals much more than could have been illustrated by plotting the snow metrics against elevation and examining the outliers. Although the wavelet analysis is rather preliminary and descriptive, it presents results and so should be moved to the Results section.
**AC (2):** Following your recommendation, in new manuscript we have moved the wavelet analysis in the Results section (See Section 3.4 from page 17 to page 19).

**RC (3):** 15
The difference between "snow cover duration" and "number of days with snow" is not clear, and is not made clear until line 183. State "number of days with snowfall" throughout.
**AC (3):** Ok, thank you for the suggestion. We have replaced "number of days with snow" with "number of days with snowfall". Moreover, in the revised version of the manuscript, we have provided a general definition of such variables in the introduction section (See pag. 2, Lines 56-59).

**RC (4):** 31
Snowfall is certainly an essential climate variable, but it is not a GCOS Essential Climate Variable distinct from precipitation, so do not use that specific term.
**AC (4):** Ok, in the new manuscript version we have removed this term.

**RC (5):** 257-300

The description of Climatol tests is barely comprehensible without reading the references.

**AC (5):** Following this suggestion, we have revised the description of Climatol test (See pag. 8-10, Lines 261-311).

**RC (6):** 360

Relationships of PCs to geographical features are stated but not made clear to the reader.

**AC (6):** Thank you for this comment. Here we provide a detailed description of the Principal Component Analysis (PCA) results for each of the three investigated variables: snow cover duration (SCD), number of days with snowfall (NDS) and height of new snow (HN). Such analysis has been included in the new version of the manuscript as Appendix B (See pag. 26-32, Lines 794-882).

It is important highlighting that the aim of our PCA analysis is to identify the dominant recurring spatial patterns over time of the investigated parameters. A natural evolution of this type of analysis is the research of the atmospheric circulation characteristics that are associated with the identified spatial structures of SCD, NDS and HN variables. This aspect is very interesting and fits well with our research interests. However, it falls out the scope of this paper, so it will be addressed in future work.

For SCD, we have selected the first four Principal Components (PCs), which account for the 75% of the total variance. Fig. 1 of this document shows the spatial pattern of the PC scores. Please consider Fig. 1 of the manuscript for locations mentioned therein. The first PC (Fig. 1a), 
[revised manuscript text omitted]

**RC (7):** 360
There is some appeal to having elevation on the y-axis, but it would conventional for it to be on the x-axis as the independent variable. This would also better show the overlap in elevation between clusters and the increasing gradient. Rather than the generic x = ay^b, it would be better to show the power fit equations as SCD = az^b.

**AC (7):** Following the valuable referee's suggestion, we have revised the Fig. 5, Fig. 6 and Fig. 7 of our manuscript. Here we present the new version of such figures.

[Figure]

**Figure 5**: Climatology of snow cover duration (SCD) for (a) full, (b) early, (c) winter and (d) late season. Average values are for the period 1971-2000. Each point represents a station that is color-coded according to the membership cluster. The black solid line represents the power fit. The text boxes show the power fit equation and the average and standard deviation values for each cluster.

[Figure]

**Figure 6:** Climatology of number of days with snow (NDS) for (a) full, (b) early, (c) winter and (d) late season. Average values are for the period 1971-2000. Each point represents a station that is color-coded according to the membership cluster. The black solid 1095 line represents the power fit. The text boxes show the power fit equation and the average and standard deviation values for each cluster.

[Figure]

**Figure 7:** Climatology of height of new snow (HN) for (a) full, (b) early, (c) winter and (d) late season. Average values are for the period 1971-2000. Each point represents a station that is color-coded according to the membership cluster. The black solid line represents the power fit. The text boxes show the power fit equation and the average and standard deviation values for each cluster.

**RC (8):** 464
"subset" would be a more widely comprehensible term than "aliquot".
**AC (8):** Ok, thank you.

**RC (9):** Figure 11
Does this contradict recovery of NDS in the Southern Apennines cited in the introduction?
**AC (9):** According to Capozzi et al. (2022) and Anella et al. (2023), a recovery in NDS has been observed, in the Southern Apennine area, in the last 20 years (i.e. in the 2000-2020 period). Note that this evidence emerged from the analysis of Montevergine time series, which is one of the few historical series that extends up to recent years. In our study, for the reason explained in Section 2.2, we focused on the 1951-2001 period. Therefore, the results sketched in Figure 11 are not in contradiction with the NDS recovery mentioned in the Introduction (i.e. the recovery occurred after 2000).

**RC (10):** 593
XX century?
**AC (10):** Yes, XX century, sorry for the mistake.

**RC (11):** Figures 12 and 13
The captions should state that arrows pointing to the right indicate that signals are in phase.
**AC (11):** Ok, Done. Thank you.

**RC (12):** 675
When claiming 90% confidence, it would make more sense to quote the 90% confidence interval.
**AC (12):** Ok, Done. Thank you for this suggestion.

**REVIEWER #2**

**RC:** Dear Authors, dear Editor,
Thank you for proposing this study. The topic is interesting and timely, the methods used are sound, and the focus on the Appenine area is a valuable complement to pre-existing work and snow data analyses.
Therefore I find the presented work very valuable and worth publishing - but naturally, I have some suggestions to try to improve it.
**AC:** Dear Referee, we are very grateful for the positive evaluation of our manuscript and for the comments and the suggestions, which help us to improve our paper. Here we provide a point-by-point response to his/her comments. Please note that all required changes have been marked in yellow in the new manuscript version.

**RC (1):** I join referee one in his/her concern that the data should be made openly accessible (for instance via a doi associated to the present paper) to comply with Copernicus guidelines.
**AC (1):** Following this valuable suggestion, we have deposited the dataset that supports this study in the Zenodo open data repository (CERN). The dataset can be accessed through the following link: https://zenodo.org/records/12699507.
We have accordingly modified the Code/Data Availability Section of the manuscript (See Pag. 32, Lines 887-888).

**RC (2):** As also assessed by referee 1, "Number of days with Snow" / NDS is too vague (notably L55) and the description comes too late in the paper. As I understand from L 183 it could be formulated as Number of days with fresh snow accumulation on the ground.
**AC (2):** In the new manuscript version, "Number of days with snow" has been replaced by "Number of days with snowfall". In addition, following your valuable comment, in the revised version of the manuscript we have provided a general definition of such variables in the introduction section (See pag. 2, Lines 56-59).

**RC (3):** The Standard Normal Homogeneity Test procedure is barely understandable the way it is currently presented without reading further reference. I suggest to explain the general philosophy underlying the test.
**AC (3):** Thank you for the suggestion. We have added the following brief description (See pag.9, Lines 282-298). Note that the changes with respect to the original manuscript version are highlighted in yellow.

"Climatol has been employed in this study also to check for homogeneity of the investigated time series. The use of this toolbox for the homogenisation of snowfall data has been explored, with encouraging results, in some recent works (Buchmann et al., 2022; Buchmann et al., 2023). As described in detail by Guijarro (2018) and by Kuya et al. (2022), the Climatol homogenization method is based on the Standard Normal Homogeneity Test (SNHT; Alexandersson, 1986) for the identification of the breaks and on a linear regression approach for the adjustments (Easterling and Peterson, 1995). The SNHT falls within homogenization procedures that are able to identify an inhomogeneity without knowing a priori the time of the break point in the time series and that can also estimate the magnitude of the detected break. The basic idea underlying this method consists in using neighbouring stations as a reference to identify inhomogeneities in the station being tested (the candidate station). Such assumption requires the existence of a sufficient correlation level between test and reference stations. More specifically, SNHT uses normalised series of the ratios/differences

(hereafter, $Q$) between e.g. precipitation/temperature at candidate station and neighbouring reference stations. The test is based on the null hypothesis that the $Q$ series has a constant mean level, i.e. that the candidate series is homogeneous, and the alternative hypothesis that the mean level of the $Q$ series changes abruptly from one level to another at some time. For each point of the time series, a test value, based on a comparison between the means of the two subsamples before and after the potential breakpoint, is computed as described in detail in Alexandersson and Moberg (1997). The null hypothesis is rejected if the maximum test value of all dividing points in the $Q$ series is greater than a predefined critical level. In Climatol, the SNHT is applied to the anomalies time series previously introduced in the description of the tolerance test."

**References**

Alexandersson, H., & Moberg, A. (1997). Homogenization of Swedish temperature data. Part I: Homogeneity test for linear trends. International Journal of Climatology: A Journal of the Royal Meteorological Society, 17(1), 25-34.

**RC (4):** More generally, the use of the terms "snow" is sometimes misleading throughout the paper, as illustrated in the expression "number of days with snow" . We don't know wether this is atmospheric snow (snowfall) or ground covering snow. Please be more specific.

**AC (4):** Ok, thank you for the suggestion. As stated in the reply to RC (2), in the new manuscript version "Number of days with snow" has been replaced by "Number of days with snowfall". In addition, following your valuable comment, in the revised version of the manuscript we have provided a general definition of snow cover duration and number of days with snowfall in the introduction section (See pag. 2, Lines 55-58).

**RC (5):** Both the title and the abstract draw the focus on snow *precipitation* or *snowfall*. However, based on the variables analyzed (snow cover duration, number of days with snow, total height of new snow) the focus is at least equally on snow on the ground as on snowfall. This should be revised in order to convey a more precise message.

**AC (5):** Thank you for this valuable suggestion. To avoid ambiguity, we have revised the title of our manuscript, replacing "snowfall" with "snow". The term "snow" may be considered more general and inclusive of different types of data (e.g. snow cover, snow precipitation amount and snow frequency of occurrence) than "snowfall". As an example, Scherrer et al. (2013) have used a similar title ("Snow variability in the Swiss Alps, 1864-2009"), in a work that considered different snow indicators, such as new snow sums, maximum new snow and days with snowfall.

Therefore, in the new manuscript version we have used the term "snow" to generally mention all the snow variables employed in our study and we will replace it with "snowfall" when referring a specific parameter, such as the "Number of days with snowfall".

**AC (7):** Ok, thank you for the suggestion. In the new manuscript version, we have modify it (See pag. 6, Line 176).

**RC (8):** In one of the authors' previous work mentioned in the Introduction, a recent recovery of snow cover duration, HN and NDS is mentioned in the Southern Appenines (L 67 68). Unfortunately the present study ends by 2000 while the recovery at Montevergine Observatory appears after 2000. Would it be possible to bridge the gap between both studies and mention this in the discussion, enlightening the dependence of trends to timeframe/period length, and also the connection with AO/NA that was seen in this previous study but found not relevant in the present one?

**AC (8):** Ok, thank you for this suggestion. In the Discussion section, before starting the comparison between the results of our study and previous literature, we state that "linear trends magnitude and their statistical significance are strongly dependent from the analysed time window" (See pag. 20, Line 625). For such reason, we focused on previous papers that considered periods having a good overlap with the present work.

Regarding the connections with Arctic Oscillation (AO), following your suggestion, we have mentioned that previous studies on the Apennine region highlighted a recovery in snow cover duration, snowfall amounts and number of days with snowfall after 2000 and that this rebound in snow indicators is closely linked to the AO. From the preliminary results presented in this work, based on the Cross Wavelet Transform, it emerged that AO exerts a less relevant influence than EMP on the nivometric regime of the investigated Apennine region. However, we feel that additional analyses are necessary to better assess the relationships between this important atmospheric mode and the snow variability in the study area. It is possible to assume that non-negligible difference might exist between western and eastern sectors of the Apennines (the first ones might be more "sensitive" to the AO variability). We have added such remarks to the Discussion section (See pag. 21, Lines 667-673).

Edits :

**RC (9):** L61 : clear -> clearly
**AC (9):** Ok, Done. Thank you.

**RC (10):** L 106 : ad -> an
**AC (10):** Ok, Done. Thank you.

**RC (11):** L 115 : southwest -> south east
**AC (11):**  Ok, Done. Thank you.

**RC (12):** L 223 : quote -> quite
**AC (12):** Ok, Done. Thank you.

**RC (13):** L 263 : use -> uses
**AC (13):** Ok, Done. Thank you.

**RC (14):** L 355 : scree -> screen
**AC (14):** "Scree plot" is correct, because it refers to a common method for determining the number of Principal Components to be retained.
(See for example https://www.sciencedirect.com/topics/mathematics/scree-plot)

**RC (15):** L 511 : second "in" -> and
**AC (15):** Ok, Done. Thank you.

**RC (16):** L 539 one word is missing (likely "to" before "contextualize").
**AC (16):** We are sorry for the error. We have added the missing word.

**RC (17):** L 559 : stations -> station
**AC (17):** Ok, Done. Thank you.

**RC (18):** L 689 : means -> by means
**AC (18):** Ok, Done. Thank you.